# Systematic perturbation of retroviral LTRs reveals widespread long-range effects on human gene regulation

Daniel R Fuentes[1,2], Tomek Swigut[2], Joanna Wysocka[2,3,4]*

[1]Cancer Biology Program, Stanford University School of Medicine, Stanford, United States; [2]Department of Chemical and Systems Biology, Stanford University School of Medicine, Stanford, United States; [3]Department of Developmental Biology, Stanford University School of Medicine, Stanford, United States; [4]Howard Hughes Medical Institute, Stanford University School of Medicine, Stanford, United States

**Abstract** Recent work suggests extensive adaptation of transposable elements (TEs) for host gene regulation. However, high numbers of integrations typical of TEs, coupled with sequence divergence within families, have made systematic interrogation of the regulatory contributions of TEs challenging. Here, we employ CARGO, our recent method for CRISPR gRNA multiplexing, to facilitate targeting of LTR5HS, an ape-specific class of HERVK (HML-2) LTRs that is active during early development and present in ~700 copies throughout the human genome. We combine CARGO with CRISPR activation or interference to, respectively, induce or silence LTR5HS en masse, and demonstrate that this system robustly targets the vast majority of LTR5HS insertions. Remarkably, activation/silencing of LTR5HS is associated with reciprocal up- and down-regulation of hundreds of human genes. These effects require the presence of retroviral sequences, but occur over long genomic distances, consistent with a pervasive function of LTR5HS elements as early embryonic enhancers in apes.
DOI: https://doi.org/10.7554/eLife.35989.001

*For correspondence:
wysocka@stanford.edu

Competing interests: The authors declare that no competing interests exist.

## Introduction

Nearly half of the human genome is composed of transposable elements (TEs), which are increasingly being recognized not just as parasitic DNA, but as an important source of regulatory innovation for the host (*Chuong et al., 2017*; *Feschotte, 2008*; *Rayan et al., 2016*; *Thompson et al., 2016*). In particular, endogenous retroviruses (ERVs), which comprise about 8% of the human genome, are sequences derived from ancient retroviruses whose germ-line infections have persisted through millions of years of evolution (*Feschotte and Gilbert, 2012*; *Johnson, 2015*; *International Human Genome Sequencing Consortium et al., 2001*). At the time of endogenization, ERVs, like all retroviruses, contain 5' and 3' long terminal repeats (LTRs) that flank open reading frames encoding retroviral proteins; over time, these LTRs accumulate mutations and often undergo homologous recombination, which reduces them to so-called 'solo' LTRs (*Greenwood et al., 2018*; *Kassiotis and Stoye, 2016*; *Stoye, 2012*; *Young et al., 2013*). In their capacity as retroviral promoters, LTRs are enriched for transcription factor motifs and thus are a particularly fertile substrate for evolving new regulatory elements that can be exapted for host gene regulation. Many examples of such exaptations now exist, for example: in the mouse two-cell (2C) stage embryo, MERVL elements serve as alternative promoters for a subset of mouse 2C genes (*Macfarlan et al., 2012*), while LTRs of a human ERV, MER41, can function as interferon-inducible enhancers (*Chuong et al., 2016*). Epigenomic mapping studies detected cell type-selective active enhancer signatures at thousands of LTRs, suggesting that acquisition of tissue-specific or inducible regulatory functions by these elements is a

widespread phenomenon that may have profound effects on host gene regulatory networks (*Bourque et al., 2008*; *Chuong et al., 2013*; *Huda et al., 2010*; *Kunarso et al., 2010*; *Martens et al., 2005*; *Sundaram et al., 2014*; *Thurman et al., 2012*; *Trizzino et al., 2017*; *Jiang et al., 2014*; *Wang et al., 2012*). Furthermore, emerging evidence suggests that a large proportion of primate-specific enhancer/promoter sequences, as well as those that changed their activity most recently, since the separation of humans from chimpanzees, originate from TEs (*Jacques et al., 2013*; *Prescott et al., 2015*; *Rayan et al., 2016*; *Trizzino et al., 2017*). Thus, understanding the functional impact of TEs on gene regulation is essential for comprehending the emergence of primate- and human-specific traits.

Despite evidence suggesting the importance of LTRs and other TEs in rewiring gene regulatory networks, most current studies are either correlative or focus on the analysis of individual insertions, rather than on systematically perturbing specific TE classes, with one notable exception of a report utilizing transcription activator-like effector (TALE) fused to effector domains for functional perturbations of mouse LINE1 elements (*Jachowicz et al., 2017*). This knowledge gap is associated with technical challenges, as LTR subfamilies are often present in hundreds or thousands of copies, which are highly repetitive, but, due to accumulated mutations, sufficiently sequence-divergent to prevent their recognition by a single short-sequence-dependent factor, such as a zinc finger protein or CRISPR guide RNA (gRNA). To overcome these limitations and develop a strategy for systematic interrogation of TE function, we leveraged our recently developed method for gRNA multiplexing called CARGO (Chimeric Array of gRNA Oligos), which allows for the introduction of tens of gRNAs into single cells (*Gu et al., 2018*).

Here, we couple CARGO with nuclease dead Cas9 (dCas9) fused to an activation or repression domain (CRISPRa and CRISPRi, respectively) (*Chavez et al., 2015*; *Gilbert et al., 2013*) to facilitate transcriptional induction or silencing of HERVK LTR5HS elements en masse. Among human ERVs, HERVK (HML-2) is of particular interest, as it is the most recently endogenized retrovirus, which infected the primate lineage both before and after the human-chimpanzee divergence and retained many intact proviruses with coding potential (*Barbulescu et al., 1999*; *Belshaw et al., 2004*; *Medstrand and Mager, 1998*). This ERV class contains integrations so recent that polymorphic insertions across the human population exist (*Belshaw et al., 2005*; *Shin et al., 2013*; *Wildschutte et al., 2016*). All human-specific and human-polymorphic HERVK insertions are associated with a specific LTR5 family subclass, LTR5HS, present in 697 copies in the human genome (hg38 assembly) (*Hanke et al., 2016*; *Subramanian et al., 2011*). We recently showed that HERVK is transcriptionally activated in human preimplantation embryos and in naïve, but not primed, human embryonic stem cells (hESCs) (*Grow et al., 2015*). Naïve hESCs model an early, preimplantation stage of the human blastocyst, characterized by global DNA hypomethylation similar to that observed in the inner cell mass, with transcriptional profiles and epigenetic landscapes different from those of primed hESCs, which are most similar to a later, postimplantation stage of the blastocyst (*Bates and Silva, 2017*; *Zimmerlin et al., 2017*). Embryonic activation of HERVK can also be modeled in human embryonal carcinoma NCCIT cells, which exhibit both pluripotent and tumorigenic characteristics, but, unlike naïve hESCs, are easy to maintain and manipulate. Similarly to naïve hESCs and preimplantation embryos, NCCIT cells express pluripotency transcription factors and are characterized by DNA hypomethylation and high expression of HERVK-derived transcripts and proteins (*Boller et al., 1993*; *Grow et al., 2015*; *Herbst et al., 1996*). Transcriptional reactivation of HERVK in NCCIT is associated with the acquisition of enhancer-like chromatin signatures at LTR5HS elements, raising the possibility that these elements may influence host gene expression programs (*Grow et al., 2015*).

We now demonstrate that a CARGO-based CRISPRa/CRISPRi strategy facilitates robust and specific targeting of dCas9 to ~90% of LTR5HS elements throughout the human genome for efficient activation or repression of HERVK transcripts and proteins. Moreover, perturbation of LTR5HS function by recruitment of an activator or a repressor leads to the reciprocal up- and down-regulation of nearly 300 human genes, along with widespread effects on the chromatin landscape surrounding the promoters of these genes and LTR5HS insertions. Remarkably, these effects on host gene expression occur over long genomic ranges, indicating that LTR5HS elements function as distal enhancers for a substantial number of genes. In agreement, deletion of select LTR5HS elements confirms their strong contribution to host gene transcription. These LTR5HS-regulated genes are preferentially expressed in naïve relative to primed hESCs and their transcripts are also elevated in

developing human blastocysts as compared to those of rhesus macaque, a primate species that does not contain LTR5HS insertions. These observations suggest that recent HERVK endogenization has contributed to the establishment of unique gene expression patterns in preimplantation embryos of humans and other apes. Altogether, our work provides a novel and broadly applicable strategy for functional manipulation of specific TE classes across the genome and supports a pervasive role of LTRs as embryonic gene enhancers.

## Results

### CARGO-CRISPRa/CRISPRi system for manipulating function of transposable elements across the genome

To investigate the role of HERVK LTR5HS insertions in the regulation of embryonic gene expression and, more broadly, to establish a proof of principle for using CARGO to simultaneously target hundreds of repetitive elements interspersed across the genome, we designed a CARGO array with 12 distinct gRNA transcriptional units, altogether predicted to recognize ~91% (635/697) of LTR5HS integrations in the human genome (hg38 assembly) when allowing zero mismatches between gRNA sequences and LTR5HS sequences (*Figure 1—figure supplement 1*). We computationally predict that many insertions are recognized by multiple gRNAs, with a maximum of nine gRNAs expected to target any single insertion. For example, at zero mismatches, ~87% of LTR5HS insertions are targeted by at least two gRNAs, and ~57% by at least four gRNAs (*Figure 1—figure supplement 1*), an important consideration given that a single gRNA is often insufficient for robust gene activation/silencing by CRISPRa/CRISPRi (*Cheng et al., 2013*; *Perez-Pinera et al., 2013*).

Although our custom scoring algorithm penalized potential gRNAs that target genomic regions other than LTR5HS, we 'masked' the highly related (~88% sequence similarity) HERVK LTR5A and LTR5B sequences to exclude them from negatively affecting candidate gRNA scores. Consequently, our CARGO array is computationally predicted to exhibit some binding to LTR5A and LTR5B, but should not target other classes of LTRs or other TEs (*Figure 1—figure supplement 1*), including the SVA elements, which are in part derived from the LTR5 sequence (*Hancks and Kazazian, 2010*; *Ono et al., 1987*). With this strategy, we expect 58% (178/306) of LTR5A and 50% (235/472) of LTR5B insertions to be bound when no mismatches are allowed.

We assembled CARGO LTR5HS-targeting arrays using either the *Streptococcus pyogenes* gRNA scaffold (hereafter called LTR5HS Sp) or the *Staphylococcus aureus* gRNA scaffold (LTR5HS Sa). As a non-targeting control, we also assembled a CARGO array with gRNAs that should not pair anywhere in the human genome, with the *S. pyogenes* gRNA scaffold (nontarget Sp). To couple CARGO with CRISPRa/CRISPRi approaches for systematic perturbation of function, we used the human embryonal carcinoma NCCIT model to generate six transgenic cell lines, each expressing one of the three aforementioned CARGO arrays and a doxycycline-inducible *S. pyogenes* dCas9 fused to either the strong transactivation domain VPR (dCas9-VPR; CRISPRa) or to a repressive KRAB domain (dCas9-KRAB; CRISPRi) (*Chavez et al., 2015*; *Gilbert et al., 2013*) (*Figure 1A*). Only cells expressing the LTR5HS Sp array will recruit dCas9 fusion proteins to the target regions, for either activation (dCas9-VPR) or repression (dCas9-KRAB) of HERVK/LTR5HS transcription (*Figure 1B*). By contrast, LTR5HS Sa gRNAs will not complex with the *S. pyogenes* dCas9, and thus cells with the LTR5HS Sa array serve as a control for overexpression of LTR5HS-derived short RNAs. Finally, in nontarget Sp cell lines, the gRNAs will form a complex with dCas9, but will not bind the genome (at least not in a sequence-dependent manner), thereby serving as a control for the presence of RNA-loaded dCas9 complexes (*Figure 1B*).

We next induced expression of the respective dCas9 fusion proteins with doxycycline in all six NCCIT cell lines, and assayed expression of LTR5HS-driven transcripts using RT-qPCR (*Figure 1C*). While most of the LTR5HS elements in the genome exist as solo LTRs, a subset remains associated with protein-encoding proviral sequences. We therefore also examined expression of HERVK transcripts encoding *env*, *gag*, and *pro*, as well as protein levels of Env. We found that although HERVK is already highly expressed in NCCIT cells, levels of both HERVK proviral transcripts and LTR5HS-derived transcripts further increase between 10- and 15-fold in the dCas9-VPR activating lines in the recruitment condition LTR5HS Sp, as compared to the control conditions LTR5HS Sa and nontarget Sp (*Figure 1C*). Conversely, in the dCas9-KRAB expressing lines, HERVK transcript expression

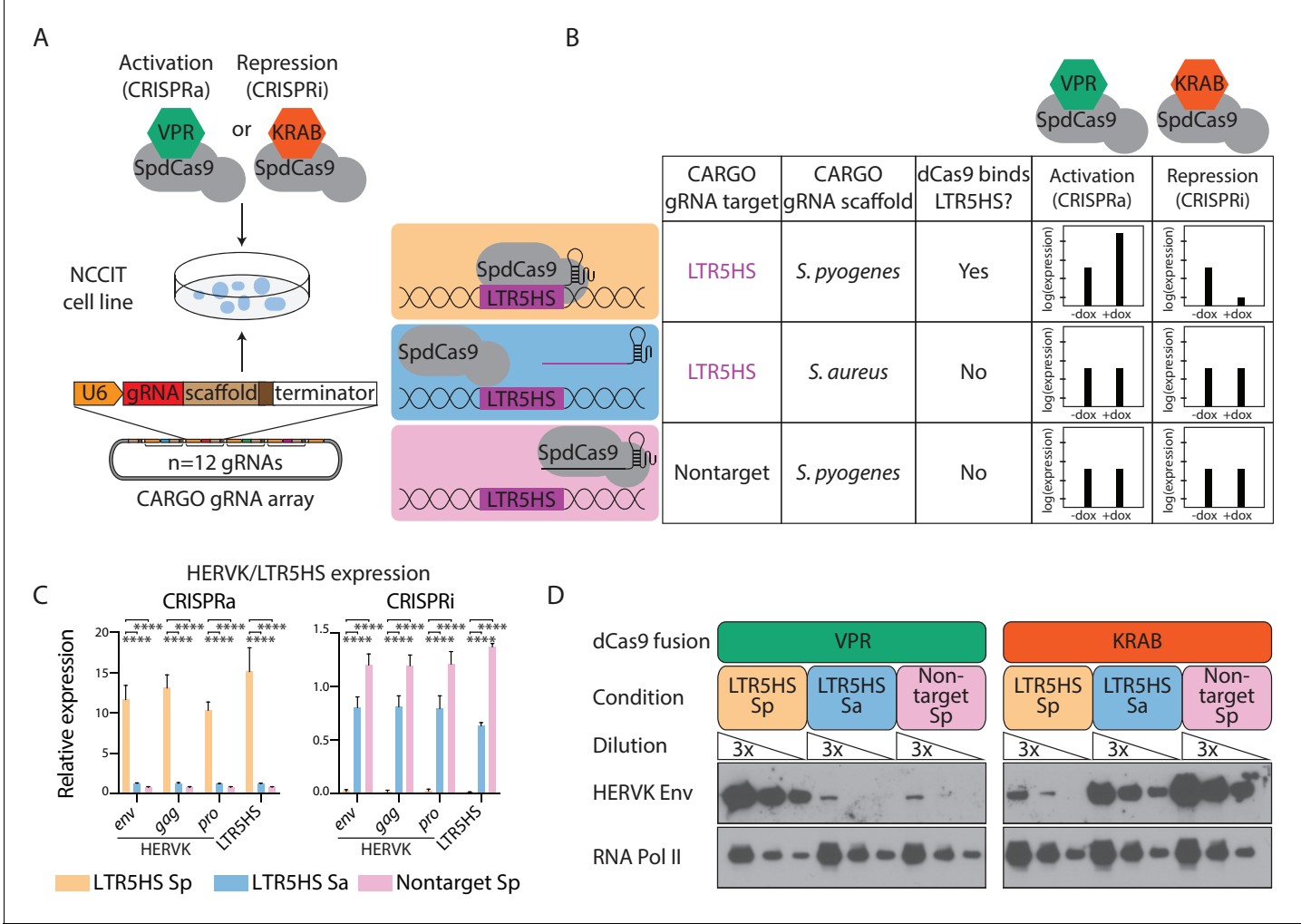

**Figure 1.** Control of HERVK/LTR5HS expression by CARGO-CRISPRa/CRISPRi. (**A**) Schematic of experimental strategy for generation of NCCIT human embryonal carcinoma cell lines expressing CARGO arrays and indicated *S. pyogenes* dCas9 fusion proteins (SpdCas9). CARGO array schematic adapted from (*Gu et al., 2018*). (**B**) Design of three CARGO arrays used in this study. CARGO arrays contain 12 distinct transcriptional units expressing gRNAs targeting LTR5HS or nontargeting gRNAs, with a scaffold sequence from the indicated bacterial species. Predicted effect of each CARGO-SpdCas9 combination on HERVK expression is shown. (**C–D**) RT-qPCR (**C**) or western blot (**D**) analysis of LTR5HS or HERVK proviral genes in NCCIT cells induced with dCas9-VPR (CRISPRa) or dCas9-KRAB (CRISPRi) and one of three CARGO arrays. In (**C**), error bars show standard deviation, and expression is shown relative to *RPL13A*, and normalized such that the average of LTR5HS Sa and nontarget Sp conditions is set to 1. ****p value < 0.0001, one-sided *t*-test. In (**D**), different exposure times have been used in left and right WB panels to allow for visualization of protein level changes upon CRISPRa and CRISPRi, respectively.

DOI: https://doi.org/10.7554/eLife.35989.002

The following figure supplement is available for figure 1:

**Figure supplement 1.** In silico binding predictions of LTR5HS targeting by gRNAs.
DOI: https://doi.org/10.7554/eLife.35989.003

decreases by over 98-fold in the binding condition LTR5HS Sp, compared to the control conditions LTR5HS Sa and nontarget Sp (*Figure 1C*). Interestingly, observed repression levels are generally as strong as or stronger than those previously reported in CRISPRi experiments with silencing of active single copy loci, attesting to the efficacy of our system. In agreement with effects on transcript expression, we also observed global increases and decreases in HERVK Env protein levels with, respectively, dCas9-VPR and dCas9-KRAB recruitment to LTR5HS (*Figure 1D*). Altogether, CARGO-CRISPRa/CRISPRi provides a robust system for manipulating the function of highly repetitive TEs such as HERVK.

## dCas9 selectively binds the majority of LTR5HS insertions

We next employed chromatin immunoprecipitation followed by high-throughput sequencing (ChIP-seq) to characterize the prevalence and specificity of dCas9 targeting to individual LTR5HS instances across the genome. We derived NCCIT lines stably expressing doxycycline-inducible dCas9 fused to EGFP (dCas9-GFP), and one of the three CARGO arrays: the recruitment (LTR5HS Sp) array or the two control (LTR5HS Sa or nontarget Sp) arrays. For each CARGO array condition, we performed ChIP-seq using three antibodies: one against Cas9, and two against GFP (example UCSC genome browser tracks are shown in *Figure 2A*). In order to avoid artifacts associated with antibody cross-reactivity, we focused our analysis on peaks called with all three antibodies. Using paired-end 150 bp sequencing allowed us to map obtained signals to individual instances of HERVK in the genome (*Figure 2—figure supplement 1*). We identified 1178 high-confidence peaks for the recruitment (LTR5HS Sp) condition, while for the control conditions we called 72 peaks (LTR5HS Sa) and 0 peaks (nontarget Sp) (*Figure 2B*), suggesting that most peaks in the LTR5HS Sp condition are due to site-specific targeting by CARGO. In agreement, the majority of dCas9 binding occurs at LTR5HS sites (591 peaks, corresponding to 85% of LTR5HS elements) or the computationally predicted and highly sequence related LTR5A/5B/5 sites (343 peaks, corresponding to 53%, 41%, and 19% of, respectively, LTR5A, LTR5B, and LTR5 instances), and is selective for the LTR5HS Sp array condition (*Figure 2B,C*). These HERVK LTR5 peaks lie almost entirely in intergenic (~68.6%) and intragenic (~30.6%) regions, with very few (~0.8%) overlapping with promoters (*Figure 2—figure supplement 2*). The remaining 244 non-LTR peaks we classify as off-targets, and these are distributed evenly between intergenic and intragenic sites. Some of these peaks (33/244, ~14%) are legitimate Watson-Crick base-pairing off-targets of the CRISPR gRNAs in the CARGO array to the human genome sequence, when allowing for up to three mismatches between gRNA and genome sequence. The rest, we believe, are simply non-specific binding sites, though some may be legitimate off-targets when permitting more than three mismatches; indeed, it is known that gRNA-to-target mismatches are tolerated beyond this threshold, especially when these mismatches are outside of the 5 bp 'seed' sequence immediately adjacent to the PAM site (*Kuscu et al., 2014*; *Wu et al., 2014*).

As would be expected, LTR5HS instances computationally predicted to align with multiple gRNAs had stronger dCas9 ChIP-seq enrichments (*Figure 2—figure supplement 3*). However, the overall correlation was only moderate (Spearman correlation coefficient $\rho = 0.57$ at zero mismatches allowed), indicating that the number of pairing gRNAs is not the sole determinant of dCas9 binding strength. Importantly, we did not observe significant binding of dCas9 to other TEs or repetitive sequences, including the SVAs (*Figure 2D*). We found only 25 individual SVA insertions (0.43% of 5750 in Repeatmasker hg38) to be bound in this experiment. Together, these data demonstrate that our CARGO-dCas9 strategy enables highly selective targeting of a specific TE class.

## Manipulation of chromatin landscape by CRISPRa/CRISPRi

We next sought to assess the effect of CRISPRa and CRISPRi on the chromatin landscape of NCCIT cells, specifically around HERVK LTR5HS sequences. To this end, we performed ChIP-seq for the histone modifications H3K27ac, H3K4me3, and H3K9me3 in wild type, parental NCCIT cells, as well as in cells expressing dCas9-VPR or dCas9-KRAB along with the LTR5HS *S. pyogenes* CARGO array (i.e. LTR5HS Sp; targeting condition). We also performed ChIP-seq for dCas9 using the Cas9 antibody described above, in the same three cell populations. As expected, in WT NCCIT cells, which do not express a dCas9 fusion, we did not detect any enrichment of dCas9 signal. We also found that in these cells, a large subset of LTR5HS elements is marked by H3K27ac and H3K4me3, with H3K4me3 showing the expected asymmetric distribution consistent with the direction of LTR-driven transcription (*Figure 3A*). Furthermore, LTR5HS insertions generally lack H3K9me3 in WT NCCIT, regardless of the presence or absence of H3K27ac and H3K4me3, suggesting that LTR5HS insertions in these cells escape KRAB-mediated repression, a major mechanism of endogenous retrovirus silencing (*Friedli and Trono, 2015*; *Rowe et al., 2010*).

Under CRISPRa and CRISPRi conditions, we detected substantial changes in all three histone marks examined (*Figure 3A*, individual LTR5HS examples are shown in *Figure 3B*). With CRISPRa, over 90% of LTR5HS elements gain a high level of H3K27 acetylation, with no appreciable change in H3K4me3. In fact, strong gains in H3K27ac occur even at those LTR5HS insertions that have low endogenous acetylation, which may suggest that ectopic enhancer activation is relatively common

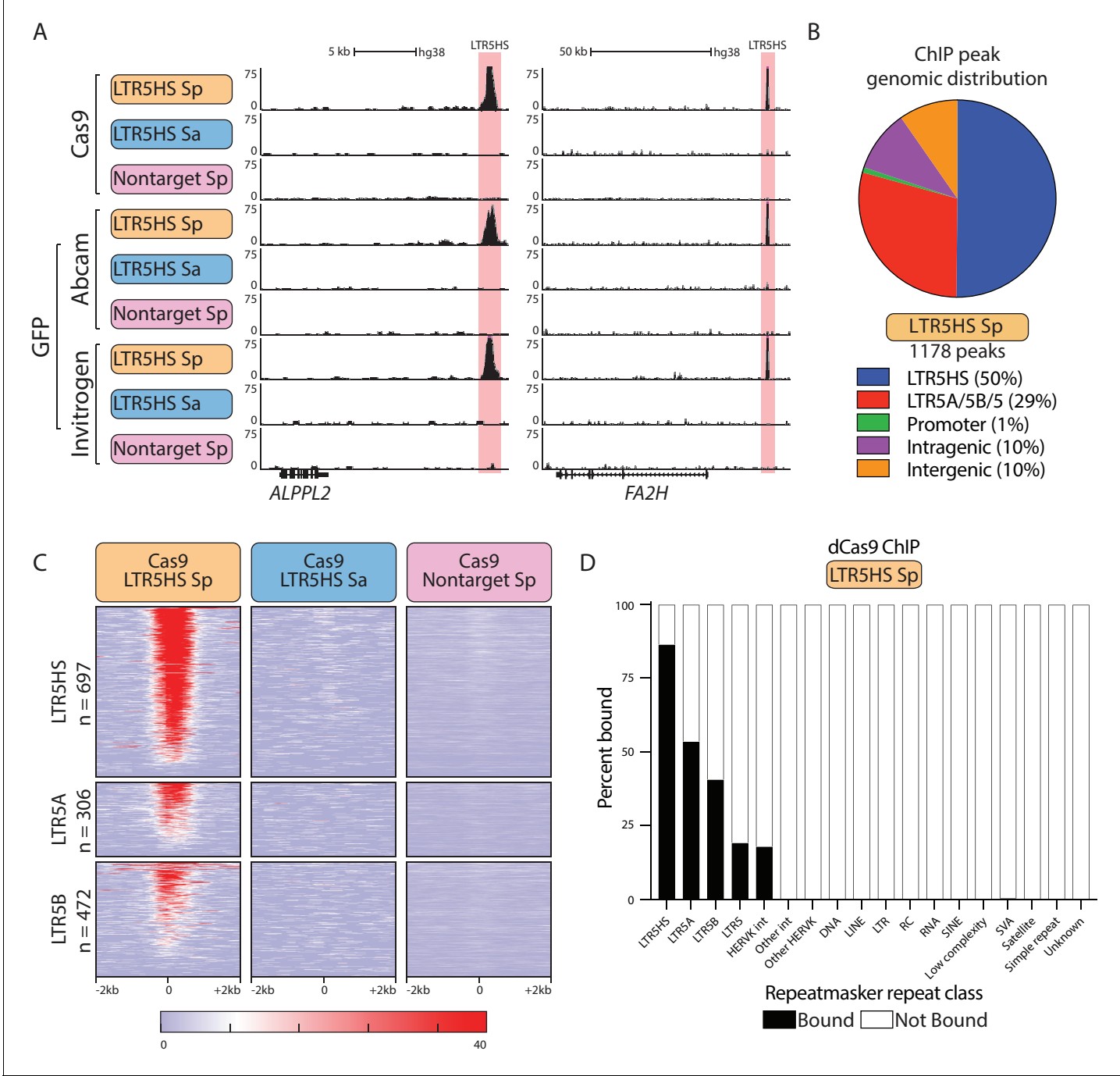

**Figure 2.** Robust and selective dCas9 targeting to LTR5HS via CARGO. (**A**) Representative UCSC hg38 genome browser tracks showing ChIP-seq profiles for dCas9 performed with three different antibodies (Cas9, GFP Abcam, GFP Invitrogen) from NCCIT cells expressing one of the three CARGO arrays (LTR5HS Sp, LTR5HS Sa, nontarget Sp; colored as in *Figure 1*). Regions around LTR5HS insertions are highlighted in pink. (**B**) Distribution of dCas9 LTR5HS ChIP-seq peaks called with all three antibodies over HERVK LTRs and known genomic features. (**C**) Heat maps of normalized ChIP-seq signal with three different CARGO arrays using Cas9 antibody. Each row represents a 4 kb window (2 kb in each direction) centered at the middle of the indicated HERVK LTR, with number of insertions of each class shown. Heat map of each LTR is sorted by Cas9 LTR5HS Sp ChIP average signal. (**D**) Percent of each Repeatmasker hg38 repeat class bound by dCas9 ChIP-seq peaks called with all three antibodies. Int, internal proviral sequences; RC, rolling circle; SVA, SINE/VNTR/Alu.

DOI: https://doi.org/10.7554/eLife.35989.004

The following figure supplements are available for figure 2:

**Figure supplement 1.** Unique mappability to LTR5HS.

*Figure 2 continued on next page*

*Figure 2 continued*

DOI: https://doi.org/10.7554/eLife.35989.005

**Figure supplement 2.** Genomic distribution of dCas9-bound HERVK LTR5.

DOI: https://doi.org/10.7554/eLife.35989.006

**Figure supplement 3.** Correlation between gRNA alignments to LTR5HS and ChIP-seq score at LTR5HS.

DOI: https://doi.org/10.7554/eLife.35989.007

and efficient with this system. Conversely, with CRISPRi, we observed a reduction in both active marks, H3K27ac and H3K4me3, and a strong concomitant increase in H3K9me3, as expected, given that KRAB repression is mediated by H3K9me3 deposition (*Figure 3A*, individual LTR5HS examples are shown in *Figure 3B*). Under both CRISPRa and CRISPRi conditions, we found strong signals of dCas9 binding, though enrichments at the corresponding elements were higher with dCas9-VPR than dCas9-KRAB (*Figure 3A*). This is likely attributable to the fact that VPR, a strong activation domain, recruits coactivators that promote nucleosomal depletion (*Calo and Wysocka, 2013*), whereas KRAB-mediated H3K9me3 facilitates chromatin compaction (*Becker et al., 2016*), which may in turn provide, respectively, positive or negative feedback for dCas9 fusion binding, especially given that nucleosomes can impede access of Cas9 to DNA (*Horlbeck et al., 2016*). Nonetheless, dCas9-KRAB still occupies and mediates H3K9me3 deposition at over 90% of LTR5HS elements (*Figure 3A*). Taken together, these data show that a large subset of LTR5HS elements is enriched in active chromatin marks in WT cells, but that targeted recruitment of dCas9 fusions results in widespread effects on the LTR5HS chromatin landscape that are consistent with the predicted activity of the fusion protein.

## Reciprocal effects of LTR5HS CRISPRa/CRISPRi on host gene expression

CARGO-CRISPRa/CRISPRi allows us to systematically test the impact of LTR5HS activation or repression on the host transcriptome. To do so, we performed RNA-seq on the six cell lines described in *Figure 1* after doxycycline induction of dCas9-VPR or dCas9-KRAB. First, we examined transcriptional changes of repetitive elements and found that, as expected, LTR5HS and HERVK transcripts are upregulated by dCas9-VPR recruitment to LTR5HS (*Figure 4—figure supplement 1A*), and downregulated by dCas9-KRAB recruitment to LTR5HS (*Figure 4—figure supplement 1B*). We next analyzed expression of non-repetitive genes, and identified 390 transcripts that significantly change in expression (false discovery rate [FDR] < 0.05) with both dCas9-VPR (CRISPRa) and dCas9-KRAB (CRISPRi) (*Figure 4A*). Of those, the majority (275 genes, 71%, *Figure 4A*, blue points in lower right quadrant) are reciprocally upregulated by CRISPRa and downregulated by CRISPRi, which is consistent both with LTR5HS-dependent regulation, and with the possibility that LTR5HS elements function as enhancers, since activation or repression of an enhancer would be expected to induce or decrease, respectively, expression of a target gene. Some genes were only affected by one of the treatments (i.e. dCas9-VPR only, 3980 genes, 1886 upregulated and 2094 downregulated, in green, or dCas9-KRAB only, 288 genes, 145 upregulated and 143 downregulated, in red, *Figure 4A*), and these effects could reflect a genuine contribution of LTR5HS to their regulation. Nonetheless, when we analyzed transcripts with respect to distance from the nearest LTR5HS, grouped by deciles from closest to furthest, we found that the majority of reciprocally affected genes (218/275, 79%) fell within the closest decile, consistent with regulation by LTR5HS (*Figure 4B*). Furthermore, the magnitude of expression changes of genes affected by CRISPRa-only or CRISPRi-only is relatively modest: only 40% and 18%, respectively, have a greater than two-fold change in expression in either direction. Most (78% and 60%, respectively) of these CRISPRa-only or CRISPRi-only affected genes fall outside of the first or second decile in distance with respect to the nearest LTR5HS (i.e. within 436 kb of the LTR5HS; compare *Figure 4A and B*), suggesting many indirect effects. In contrast, of the 275 genes reciprocally upregulated by CRISPRa and downregulated by CRISPRi, 225 (82%) show greater than two-fold change in expression in at least one condition, and 250 (91%) fall within the first or second decile of distance from the nearest LTR5HS. Therefore, we further focus on the 275 genes that show reciprocal transcriptional effects in CRISPRa/CRISPRi, and we refer to them as LTR5HS-regulated transcripts.

For these 275 LTR5HS-regulated transcripts, we found that the nearest LTR5HS insertion is upstream of the promoter in 150 cases, and downstream in 125 cases. This finding suggests that

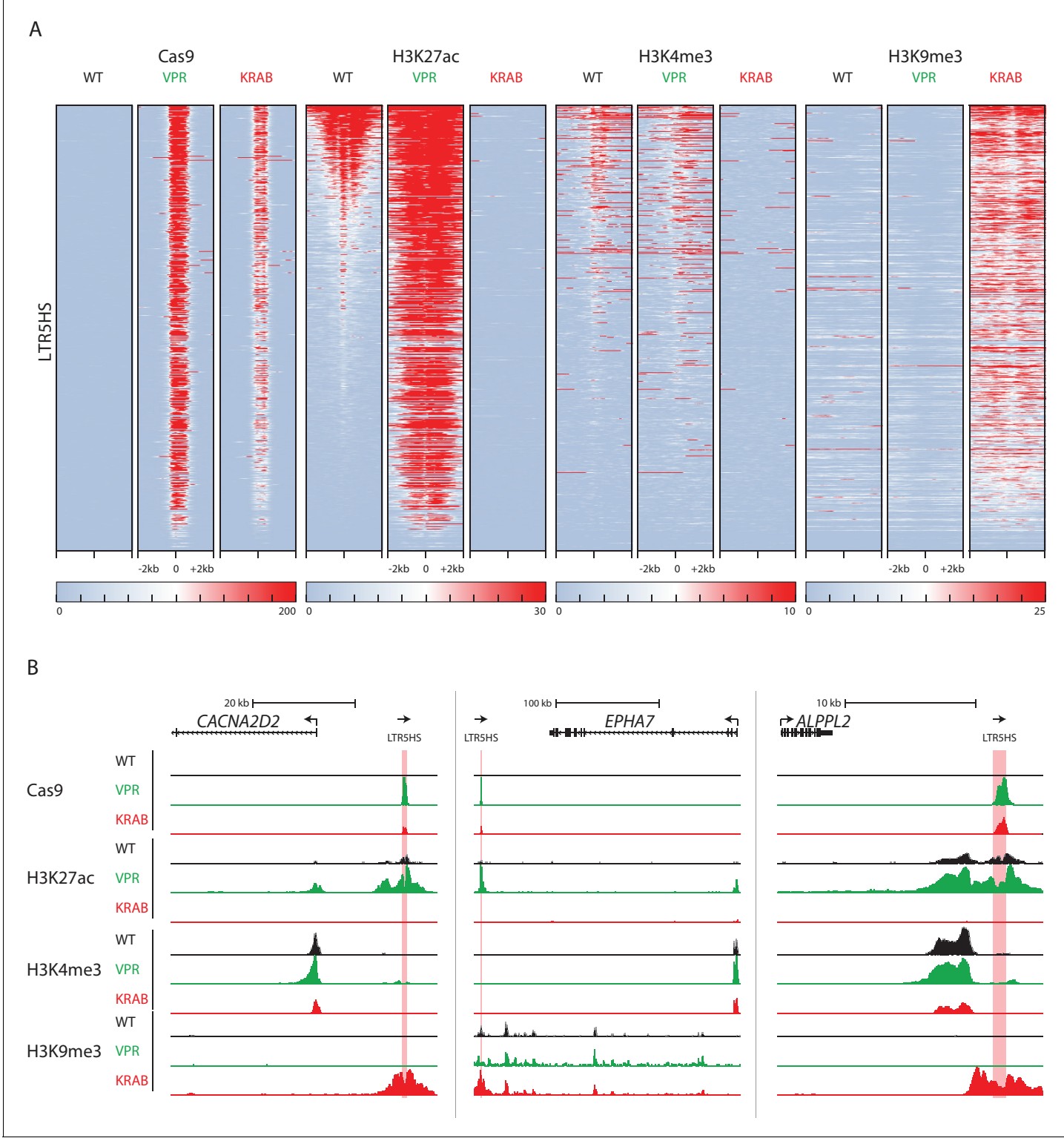

**Figure 3.** Changes in LTR5HS chromatin landscape upon CARGO-CRISPRa/CRISPRi. (**A**) Heat maps of normalized ChIP-seq signal using antibodies against Cas9, H3K27ac, H3K4me3, or H3K9me3. Heat maps for each antibody show wild type NCCIT or NCCIT cells expressing dCas9-VPR or dCas9-KRAB fusion along with LTR5HS Sp CARGO array. Each row represents a 4 kb window (2 kb in each direction) centered at the middle of HERVK LTR5HS. All heat maps are sorted by H3K27ac signal in WT NCCIT. (**B**) UCSC hg38 genome browser tracks showing ChIP-seq profiles for Cas9, H3K27ac, H3K4me3, and H3K9me3, in WT NCCIT, CRISPRa targeting condition (dCas9-VPR), and CRISPRi targeting condition (dCas9-KRAB). LTR5HS insertions are highlighted in pink. Arrows show direction of transcription of coding genes and LTR5HS elements.
DOI: https://doi.org/10.7554/eLife.35989.008

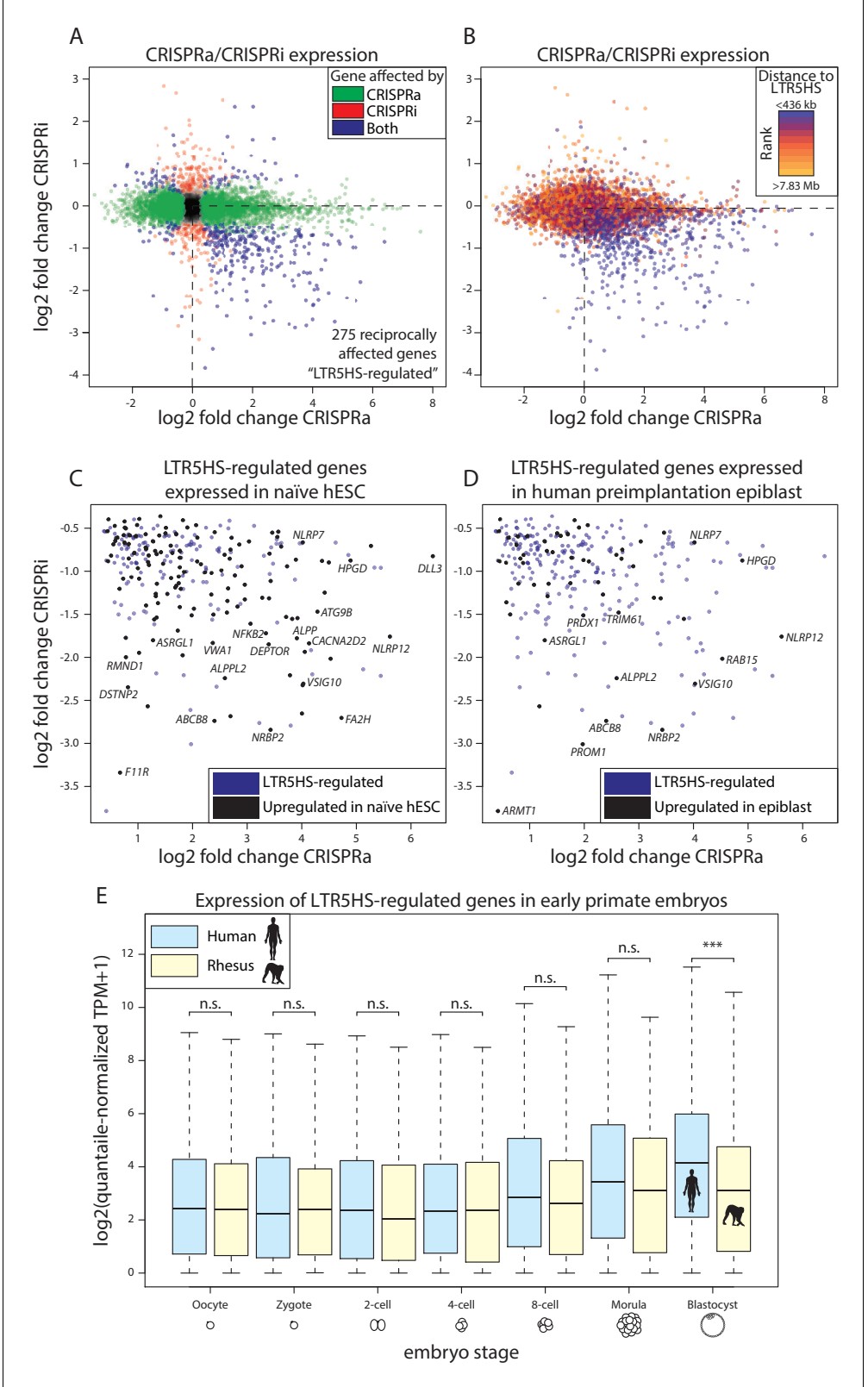

**Figure 4.** Reciprocal effects of LTR5HS CARGO-CRISPRa/CRISPRi on host gene expression. (**A**) Gene expression log2 fold change of CRISPRi (recruitment vs. control) vs. log2 fold change of CRISPRa (recruitment vs. control). Green, genes affected by CRISPRa alone; red, genes affected by CRISPRi alone; blue, genes affected by both CRISPRa and CRISPRi. Dotted line at lower right quadrant delineates LTR5HS-regulated transcripts reciprocally

*Figure 4 continued on next page*

*Figure 4 continued*

upregulated by CRISPRa and downregulated by CRISPRi. (B) Plot as in (A), with genes separated into deciles by distance from nearest LTR5HS insertion. Blue, nearest decile; orange, farthest decile. Distance bins for nearest and farthest decile are shown above and below legend, respectively. (C–D) Lower right quadrant of LTR5HS-regulated transcripts in (A), with genes significantly upregulated in (C) naïve versus primed hESC or (D) human preimplantation epiblast shown in black. Data from (*Takashima et al., 2014*; *Theunissen et al., 2016*; *Yan et al., 2013*). (E) Log2-transformed expression of LTR5HS-regulated transcripts in single cells of early human and rhesus macaque embryos at indicated stages of embryogenesis. Plots show median (center line), with interquartile range (box) and whiskers show points within 1.5x the interquartile range. ***p value < 0.001; n.s. not significant, Wilcoxon-Mann-Whitney test. Of the 275 LTR5HS-regulated transcripts, 193 are one-to-one orthologous genes between human and rhesus. Only expression of these genes was considered in this analysis.

DOI: https://doi.org/10.7554/eLife.35989.009

The following figure supplement is available for figure 4:

**Figure supplement 1.** Additional CRISPRa/CRISPRi RNA-seq analyses.

DOI: https://doi.org/10.7554/eLife.35989.010

---

even downstream LTR5HS insertions can have a transcriptional effect on the gene, meaning that these insertions do not serve as alternative promoters. Furthermore, since LTR sequences do have a natural orientation, we also examined the relative orientation of the nearest LTR5HS for each of these genes. In the 150 cases in which the nearest LTR5HS insertion is upstream of the promoter, the LTR5HS has the same orientation as the gene (both on Watson strand or both on Crick strand) 83 times, compared to 67 in the opposite orientation. In the 125 cases in which the nearest LTR5HS insertion is downstream of the promoter, the LTR5HS has the same orientation as the gene 48 times, compared to 77 in the opposite orientation. Together, these findings suggest that neither the relative position of the LTR5HS insertion to the promoter, nor its orientation, determines its ability to effect a transcriptional change on the gene in question under CRISPRa or CRISPR, consistent with the putative enhancer function. Furthermore, when we analyzed the RNA-seq data for the presence of chimeric transcripts between LTR5HS and the LTR5HS-regulated genes, we detected an appreciable level (i.e. transcripts per million [TPM] > 1) of chimeric transcription at only four of the 275 genes (specifically, *NBPF12*, *SLC4A8*, *FA2H*, and *TIMM50*). Thus, the function of LTR5HS as alternative promoters cannot broadly explain the observed regulatory effects on host gene transcription.

Gene ontology analysis of the LTR5HS-regulated transcripts did not detect strong enrichments in specific biological processes and pathways (data not shown). Interestingly, however, even though our experiments were performed in NCCIT embryonal carcinoma cells, we analyzed previously published RNA-seq data and observed statistically significant relationships between LTR5HS-regulated transcripts and differentially expressed genes in these public datasets. Specifically, we found that 138 of the 275 LTR5HS-regulated transcripts (50%, Fisher's exact test p value = $2.63 \times 10^{-26}$) are also upregulated in naïve as compared to primed hESCs (*Figure 4C*) (*Takashima et al., 2014*; *Theunissen et al., 2016*), and that 55 of these transcripts (20%, Fisher's exact test p value = $3.85 \times 10^{-21}$) are expressed in the human preimplantation epiblast (*Figure 4D*) (*Yan et al., 2013*). These observations are consistent with potential LTR5HS-dependent gene regulation in naïve hESC and preimplantation embryos, where these elements undergo transcriptional reactivation (*Grow et al., 2015*; *Theunissen et al., 2016*). We next analyzed published single cell RNA-seq data from both human and rhesus macaque early embryos, and found that LTR5HS-regulated transcripts are more highly expressed in human than rhesus preimplantation blastocysts (*Figure 4E*; Wilcoxon-Mann-Whitney p value < 0.001) (*Wang et al., 2017*; *Yan et al., 2013*). A trend towards human-specific upregulation of these transcripts can be observed starting at the 8-cell stage through the morula, although it only reaches statistical significance in the blastocyst (*Figure 4E*). Given that the rhesus genome does not contain any LTR5HS insertions, and that LTR5HS-driven expression in the developing human embryo begins at the 8-cell stage and peaks in the blastocyst (*Grow et al., 2015*), these observations suggest that the acquisition of LTR5HS after the split of apes from old world monkeys has contributed to increased expression of a subset of preimplantation genes specifically in apes. We then analyzed the evolutionary age of the LTR5HS insertions closest to the LTR5HS-regulated transcripts (ranging from over 20 million years for the oldest elements to a couple of hundred thousand years for the youngest). We observed no bias for older insertions to be

associated with regulatory changes (*Figure 4—figure supplement 1C and D*) and consequently, a subset of LTR5HS-regulated transcripts was linked to human-specific LTR5HS instances (i.e. those 5 million years old or younger). These observations raise the intriguing possibility that LTR5HS may mediate not only ape-specific, but also human-specific features of early embryonic gene regulation.

## LTR5HS activation and repression affect host gene transcription over long genomic distances

A hallmark of enhancer elements is their ability to activate host gene expression over long genomic distances and in an orientation-independent manner. We noted that although most LTR5HS-regulated transcripts fell within the closest decile category with respect to distance from the nearest LTR5HS, this category encompassed distances of up to ~436 kb (*Figure 4B*). We therefore took an LTR5HS-centric approach, and examined changes in host gene expression in relation to distance from each gene transcription start site (TSS) to the nearest LTR5HS at higher resolution within the ±200 kb domain. We found that expression of genes with promoters located not only in direct proximity of LTR5HS, but up to ~200 kb upstream or downstream of LTR5HS, was significantly upregulated by recruitment of dCas9-VPR (CRISPRa) to LTR5HS (LTR5HS Sp), compared to controls (LTR5HS Sa and nontarget Sp), but at further distances the changes became non-significant (*Figure 5A*, see *Supplementary file 1* for statistical analysis). We observed the opposite effect with recruitment of dCas9-KRAB (CRISPRi) to LTR5HS, with genes within ~200 kb upstream or downstream of LTR5HS elements, but not those further away, showing significant downregulation (*Figure 5B*, see *Supplementary file 1* for statistical analysis). Thus, activation or repression of LTR5HS can exert long-range effects on host gene transcription, in agreement with the function of these elements as long-range enhancers.

Given that many LTR5HS-regulated transcripts are also differentially expressed between naïve and primed hESC (*Figure 4C*) and that LTR5HS appears to be selectively active in naïve as compared to primed hESC (*Grow et al., 2015*), we used publicly available data from (*Theunissen et al., 2016*) and (*Takashima et al., 2014*) to probe the relationship between the distance from the LTR5HS and changes in expression between naïve and primed hESC. We observed naïve state-biased expression of genes located up to 40–120 kb away from the LTR5HS, depending on the dataset used for the analysis (*Figure 5C–D*, see *Supplementary file 1* for statistical analysis).

In contrast, we found more limited impact on transcription of genes near LTR5A and LTR5B, where only local effects can be detected (*Figure 5—figure supplement 1A–D*). Given that 53% of LTR5A regions and 41% of LTR5B regions are bound by dCas9 (*Figure 2D*), this suggests that LTR5A/B insertions likely do not have robust long-range enhancer activity in NCCIT cells, although we cannot exclude the possibility that the weaker transcriptional effects are associated with lower enrichments of dCas9 fusion proteins at these elements. Nonetheless, LTR5HS, which contains an OCT4 motif, is preferentially bound by OCT4 and p300 as compared to LTR5A and LTR5B, which do not contain the motif (*Grow et al., 2015*; *You et al., 2013*). We also analyzed publicly available ChIP-seq data and observed OCT4 and H3K27ac enrichments at LTR5HS in NCCIT and naïve hESC, but not primed hESC, while no enrichments were detected at LTR5A or LTR5B (*Figure 5—figure supplement 2*). These data suggest that genuine functional differences in regulatory capacity exist within distinct subclasses of HERVK LTR5 elements, and that their regulatory activity is cell type-specific. As a control, we analyzed gene expression changes under CRISPRa and CRISPRi conditions with respect to distance from HERVE LTR2, a class of LTR that is not targeted in these experiments. As expected, we found no effect on genes near this LTR class, confirming that the transcriptional changes observed are dependent on specific targeting of HERVK LTR5HS (*Figure 5—figure supplement 1E–F*).

We next sought to determine if transcriptional changes observed at the LTR5HS-regulated genes under CRISPRa and CRISPRi conditions are accompanied by differences in histone modifications at the promoters of these genes. We used the histone modification ChIP-seq data described above to examine patterns of H3K27ac, H3K9me3, and H3K4me3 surrounding the promoters of the 275 LTR5HS-regulated transcripts (i.e. blue points in lower right quadrant of *Figure 4A*). Most of these promoters are marked by at least some H3K27ac and H3K4me3 in WT NCCIT, and most gain or lose, respectively, H3K27 acetylation under CRISPRa or CRISPRi conditions (*Figure 5E*). Notably, these changes occur in the absence of direct dCas9 binding to the promoters, suggesting that they result from the long-range effects of LTR5HS (*Figure 5E*). Furthermore, although some gains of

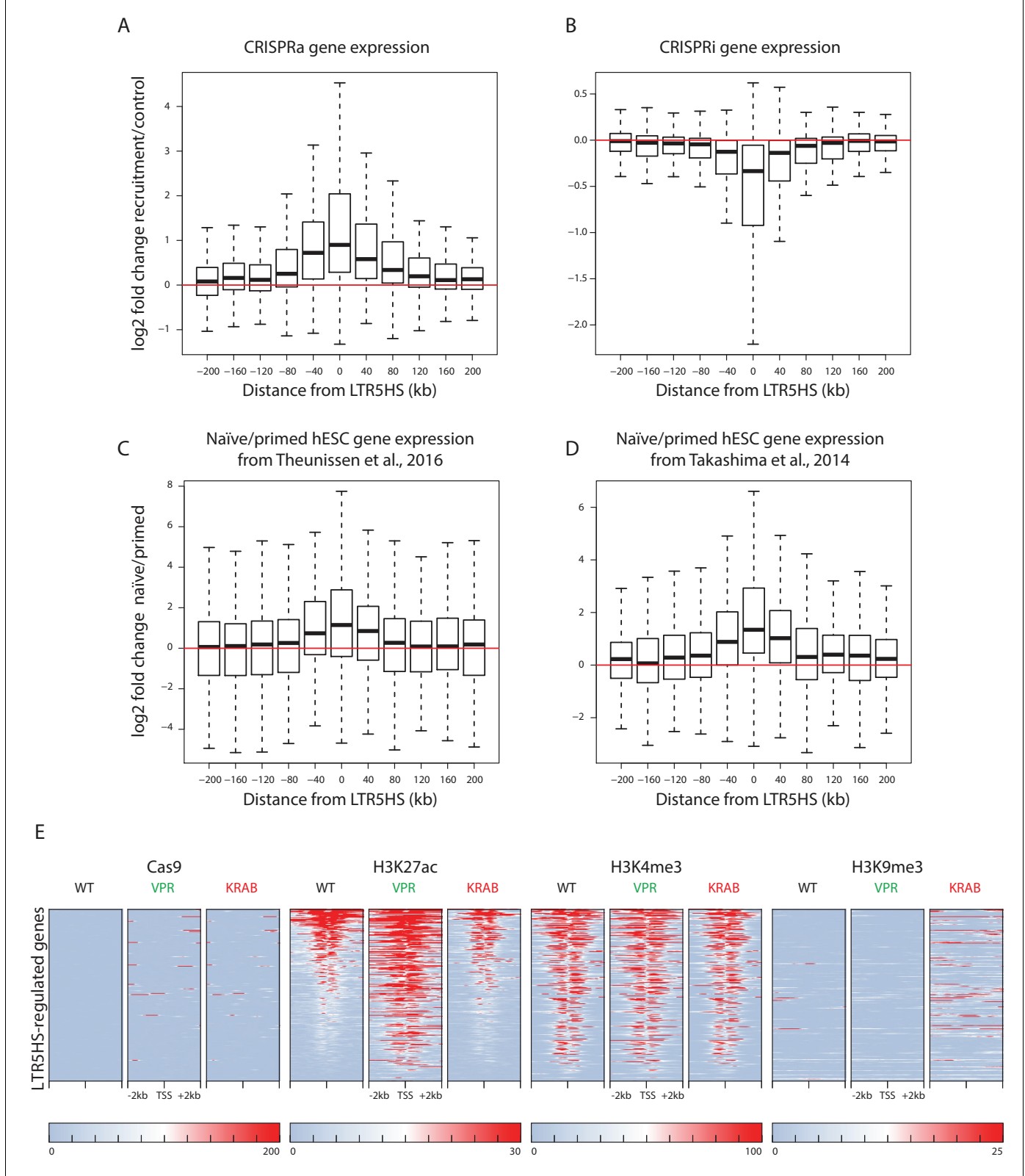

**Figure 5.** LTR5HS activation or repression affects host gene expression over long genomic distances. (**A–B**) Box plots of log2 fold change in gene expression between recruitment (LTR5HS Sp) and control (LTR5HS Sa and nontarget Sp) arrays in NCCIT cells induced with CRISPRa (**A**) or CRISPRi (**B**). (**C–D**) Box plots of log2 fold change in gene expression between naïve and primed hESC, using data from (*Theunissen et al., 2016*) (**C**) and (*Takashima et al., 2014*) (**D**). For all box plots, genes are binned into 40 kb bins centered around the indicated integer by distance from the TSS to the

*Figure 5 continued on next page*

*Figure 5 continued*

center of the nearest LTR5HS insertion. Plots show median (center line), with interquartile range (box), and whiskers show points within 1.5x the interquartile range. Statistical significance analysis of observed changes for each bin and additional bins located at distances further away from LTR5HS is presented in *Supplementary file 1*. (E) Heat maps of normalized ChIP-seq signal using antibodies against Cas9, H3K27ac, H3K4me3, or H3K9me3. Heat maps for each antibody show wild type NCCIT or NCCIT cells expressing dCas9-VPR or dCas9-KRAB fusion along with LTR5HS Sp CARGO array. Each row represents a 4 kb window (2 kb in each direction) centered around the TSS of the 275 LTR5HS-regulated genes (i.e. blue points in lower right quadrant of *Figure 4A*). All heat maps are sorted by H3K27ac signal in WT NCCIT.

DOI: https://doi.org/10.7554/eLife.35989.011

The following figure supplements are available for figure 5:

**Figure supplement 1.** Expression changes in relation to distance from LTR5A, LTR5B, and HERVE LTR2.

DOI: https://doi.org/10.7554/eLife.35989.012

**Figure supplement 2.** OCT4 and H2K27ac enrichments at HERVK LTR5 subclasses.

DOI: https://doi.org/10.7554/eLife.35989.013

**Figure supplement 3.** ChIP-seq heat maps for 275 randomly selected genes.

DOI: https://doi.org/10.7554/eLife.35989.014

H3K9me3 can be observed in the vicinity of the promoters under CRISPRi conditions, most TSS remain unmethylated at H3K9, and, unlike at the LTRs, their H3K4me3 levels are relatively unaffected, suggesting that direct silencing of promoters via H3K9me3 spreading from a nearby LTR5HS is not likely to explain the transcriptional effects we examine in this study. As a control, we performed these same analyses on a set of 275 random promoters, and we detected no changes in any histone mark under CRISPRa or CRISPRi conditions (*Figure 5—figure supplement 3*).

## Long-range effects on host gene expression are dependent on LTR5HS DNA sequence

We next sought to test whether the presence of LTR5HS DNA sequences is required for both the deposition of enhancer marks in the vicinity of the LTR5HS and for the observed long-range effects on host gene expression. To this end, we selected six genes, *CACNAD2D*, *EPHA7*, *ALPPL2*, *NFKB2*, *SERPINB9*, and *GDPD1*, that: (i) were among the 275 LTR5HS-regulated genes with reciprocal effects on expression upon CRISPRa/CRISPRi, (ii) contained no more than two LTR5HS within 1 Mb of the TSS, (iii) spanned a large range of promoter distances from LTR5HS (e.g. from ~2 kb for the closest to ~245 kb for the most distal), and (iv) represented all potential combinations of position relative to the promoter as well as orientation of LTR5HS-driven transcription with respect to the gene. We deleted the nearest LTR5HS element at each selected locus via CRISPR/Cas9 genome editing using WT NCCIT cells as a parental cell line.

We first performed ChIP-qPCR for the histone modifications H3K27ac and H3K4me1 on multiple clonal lines with or without the LTR5HS deletions at three of these loci: *CACNA2D2*, *EPHA7*, and *ALPPL2*. We found that upon deletion of the LTR5HS, both H3K27ac and H3K4me1 were significantly reduced in the regions directly flanking the LTR5HS insertion, consistent with the idea that the presence of the LTR5HS sequence is required for the deposition of these marks (*Figure 6A*). We also found that H3K27ac is significantly reduced at the promoter of two of these three genes (*EPHA7* and *ALPPL2*) upon deletion of the LTR5HS (*Figure 6—figure supplement 1*), which indicates that the presence of the distal LTR5HS sequence has a direct effect on the chromatin state of the gene's promoter.

Next, we measured the expression of each of the six genes across multiple clonal NCCIT lines with or without the deletion of the nearest LTR5HS (*Figure 6B*). For all six genes, we observed a significant decrease in expression upon deletion of the nearest LTR5HS. For *CACNA2D2*, we observed an average of ~2.6-fold decrease in expression upon deletion of the nearest LTR5HS, which is human-specific, located ~16.7 kb upstream of the TSS, and transcribed in a divergent orientation with respect to the gene (n = 6 LTR5HS deleted clones; 10 LTR5HS WT clones; *Figure 6B*). We found an average of ~8.1-fold decrease in expression after deleting the LTR5HS element closest to *EPHA7*, which is also human-specific, located ~245 kb downstream of the TSS, and transcribed in a convergent orientation towards the gene (n = 3 LTR5HS deleted clones; 15 LTR5HS WT clones). For *ALPPL2*, we measured an average of ~3.3-fold decrease in expression with deletion of the nearest LTR5HS, which is also human specific, located ~16 kb downstream of the TSS, and transcribed in the

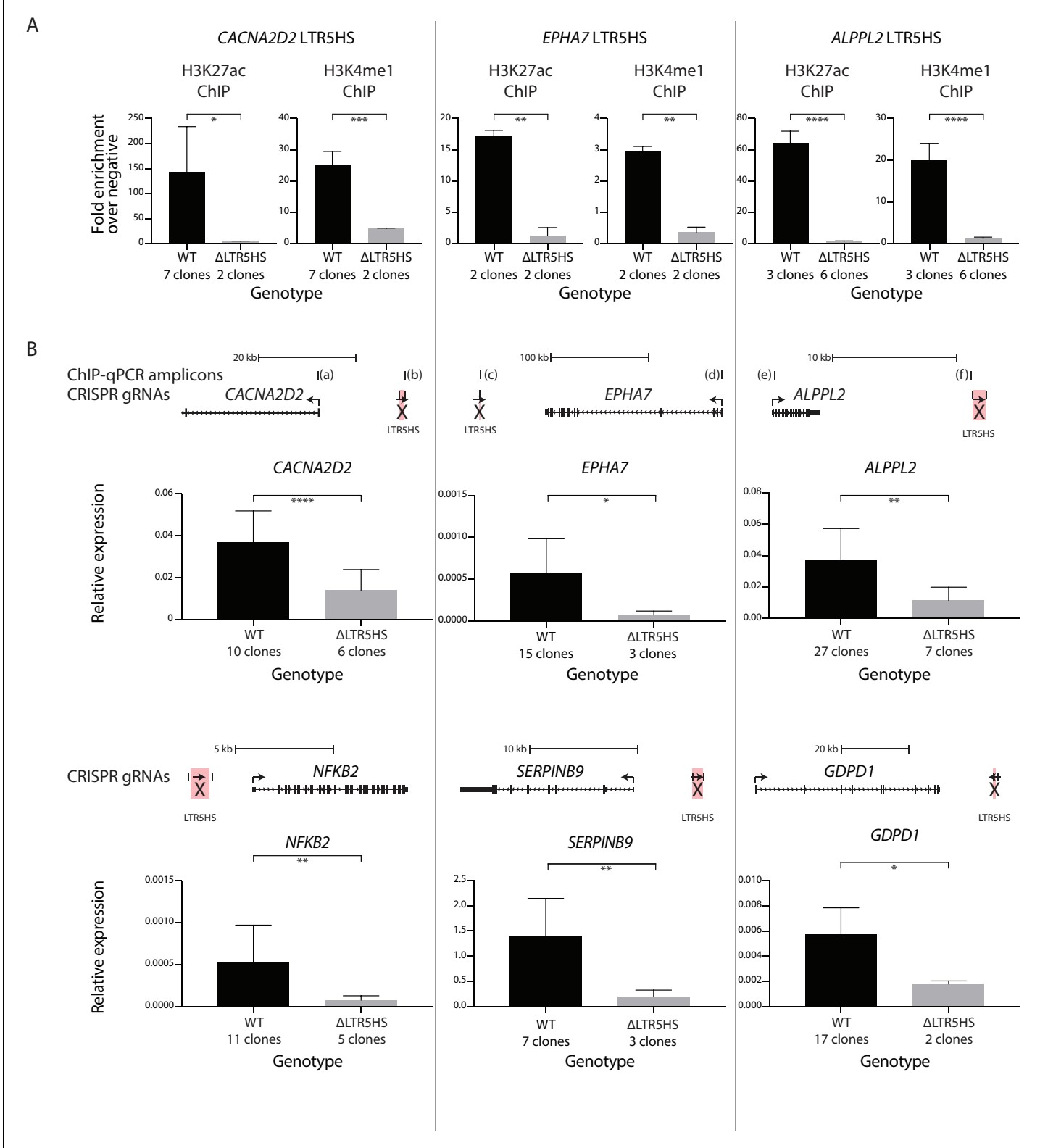

**Figure 6.** Contribution of LTR5HS sequences to chromatin marking and host gene expression. (**A**) ChIP-qPCR analysis for H3K27ac and H3K4me1 on multiple clonal lines with or without the LTR5HS deletions at indicated gene loci. Regions directly flanking the LTR5HS were analyzed for ChIP signal enrichment over two negative regions. Average signals obtained across indicated number of clones are shown. (**B**) RT-qPCR analysis of LTR5HS-regulated transcripts in multiple clonal lines with or without the LTR5HS deletions at indicated gene loci. Average expression of each gene across

*Figure 6 continued on next page*

*Figure 6 continued*

indicated number of clones is shown, measured relative to two housekeeping genes, *RPL13A* and *TBP*. Above each plot in (**B**), diagram showing TSS and nearest LTR5HS is shown to scale. Arrows show direction of transcription of coding genes and LTR5HS elements. For both (**A**) and (**B**), clones are either WT (black) or deleted for the nearest LTR5HS (LTR5HS highlighted in pink and marked with an 'X' in top panels of [B]) by CRISPR/Cas9 genome editing (gray). Error bars show standard deviation. *p value < 0.05; **p < 0.01; ***p < 0.001; ****p < 0.0001, one-sided *t*-test.

DOI: https://doi.org/10.7554/eLife.35989.015

The following figure supplement is available for figure 6:

**Figure supplement 1.** ChIP-qPCR analysis at promoters of LTR5HS-regulated genes upon deletion of nearest LTR5HS insertion.

DOI: https://doi.org/10.7554/eLife.35989.016

same orientation as the gene (both are on the Watson strand) (n = 7 LTR5HS deleted clones, 27 LTR5HS WT clones). We similarly observed an average of ~6.9-fold decrease in expression after deleting the LTR5HS element nearest to *NFKB2*, ~2.1 kb upstream of the TSS and transcribed in the same orientation as the gene (both are on the Watson strand) (n = 5 LTR5HS deleted clones; 11 LTR5HS WT clones). We found an average of ~7.0-fold loss of expression of *SERPINB9* upon deletion of the nearest LTR5HS element, ~5.8 kb upstream of the TSS and transcribed in a divergent orientation with respect to the gene (n = 3 LTR5HS deleted clones; 7 LTR5HS WT clones). Finally, upon deletion of the LTR5HS closest to *GDPD1*, ~69 kb downstream of the TSS and transcribed in a convergent orientation towards the gene, we found an average of ~3.2-fold loss of expression of the gene (n = 2 LTR5HS deleted clones, 17 LTR5HS WT clones; *Figure 6B*). These results demonstrate that long-range effects on gene regulation are directly dependent on LTR5HS DNA sequences and show that a single promoter-distal LTR can provide a very strong contribution to the overall gene activity.

## Discussion

Our study demonstrates a proof of principle for combining CARGO with CRISPRa/CRISPRi to simultaneously target hundreds of repetitive elements across the genome and manipulate their function. While we focused on HERVK LTR5HS, the strategy described here could be easily adapted to study different classes of TEs. We exploited the sequence similarity of LTR5HS insertions to target hundreds of insertions with only twelve gRNAs, with most insertions being targeted by multiple gRNAs. Given that CARGO can easily deliver 36 or more gRNAs to single cells (*Gu et al., 2018*), our approach is applicable for targeting TEs that are more sequence-divergent and/or present in higher copy numbers than LTR5HS. Furthermore, different dCas9 fusions could replace the dCas9-VPR and dCas9-KRAB fusions used in this work. These could potentially enable imaging at these loci (*Chen et al., 2013*; *Gu et al., 2018*) or local manipulation of DNA or histone modifications (*Hilton et al., 2015*; *Kearns et al., 2015*; *Lei et al., 2017*; *Liu et al., 2016*; *Vojta et al., 2016*; *Xu et al., 2016*).

The findings that CRISPRa/CRISPRi reciprocally affects expression and promoter histone modification patterns of genes located tens or even several hundreds of kilobases away from LTR5HS elements, and that CRISPR/Cas9 deletion of individual LTR5HS insertions substantially decreases expression of nearby host genes spanning a wide range of distances and distinct orientations with respect to the LTR, altogether indicate that these insertions act as enhancer elements. While multiple recent studies demonstrated the presence of enhancer chromatin signatures at various classes of LTRs, correlated them with expression of nearby genes, or directly demonstrated the importance of select individual LTR instances for host gene activity (*Chuong et al., 2013*; *Chuong et al., 2016*; *Grow et al., 2015*; *Theunissen et al., 2016*; *Thurman et al., 2012*; *Wang et al., 2014*), to our knowledge this study is the first to systematically interrogate the function of a specific LTR class in long-range gene regulation. We uncovered a broad impact of LTR5HS on host gene transcription, with 275 genes being reciprocally up- or down-regulated in our CRISPRa/CRISPRi experiments. Given the widespread redundancies in mammalian regulatory landscapes where loss of a single enhancer often has only a minor influence on expression (*Hay et al., 2016*; *Hnisz et al., 2015*; *Moorthy et al., 2017*; *Osterwalder et al., 2018*), the transcriptional effects we observe upon deletion of single LTR5HS elements are surprisingly potent, suggesting that these elements indeed function as strong and/or relatively non-redundant enhancers of their target genes.

Considering that other classes of TEs beyond LTR5HS are likely contributing to gene regulation in the early human embryo, these observations are consistent with a pervasive, rather than occasional, role of TEs in transcriptional control. In the mouse, MERVL elements in 2C stage embryos function as alternative promoters (*Macfarlan et al., 2012*), and, so far, no evidence exists to suggest that they may act as transcriptional enhancers. Hundreds of chimeric transcripts spanning junctions between 5' ERV LTRs and exons containing open reading frames were detected in these cells. However, we found no evidence of pervasive chimeric transcription between HERVK LTR5HS insertions and nearby host genes (*Grow et al., 2015* and this study), illustrating diverse mechanisms that may underlie regulatory functions in the early embryo.

Although the fact that evolutionarily young LTRs such as LTR5HS have been so extensively adapted for enhancer function may seem counterintuitive, it is important to note that preimplantation embryo cells and germ cells may be a privileged environment for such early adaptation, not only due to global DNA hypomethylation in these cells, but because in order to persist through vertical transmission, these ancient retroviruses must have been able to replicate in the germline or early embryonic cells, before the germline has been set aside. Thus, LTRs of retroviruses that successfully endogenized might have been optimized to begin with for directing expression in early embryo/germ cells. Interestingly, LTR5HS elements (but not related LTR5A/B elements) contain a consensus motif and are bound by the pluripotent stem cell/primordial germ cell/reprogramming factor and master regulator OCT4, which may have contributed both to their endogenization and cooption for enhancer function (*Grow et al., 2015* and *Figure 5—figure supplement 2*). Indeed, OCT4 plays a central role in activating pluripotency network enhancers (*Boyer et al., 2005*; *De Los Angeles et al., 2015*) and our previous work demonstrated that its binding motif is important for the ability of LTR5HS to drive transcription (*Grow et al., 2015*).

It is intriguing to consider whether regulatory repurposing of LTR5HS elements for enhancer function may have contributed to human-specific transcriptome divergence and endowed the early developmental stages of the human embryo with species-specific attributes. All LTR5HS insertions are unique to apes, and a subset is human-specific or even human-polymorphic (*Belshaw et al., 2005*; *Shin et al., 2013*; *Subramanian et al., 2011*; *Wildschutte et al., 2016*). We found that both human-specific and older, ape-specific LTR5HS elements contribute to long-range gene regulation, and that some of the genes dependent on them in embryonal carcinoma cells are also expressed in human preimplantation embryos. Interestingly, we found that transcript levels of genes that are orthologous between human and rhesus macaque and regulated by LTR5HS in human cells are significantly elevated in human blastocysts compared to rhesus blastocysts. Given that rhesus diverged from the human lineage approximately 25 million years ago (*Rhesus Macaque Genome Sequencing and Analysis Consortium, et al., 2007*), before the integration of LTR5HS, our findings suggest that a recent burst of HERVK endogenization supplied humans and other apes with new early embryonic enhancers, leading to a shift in preimplantation gene expression programs. Although there is no evidence thus far to suggest that the phenotypic consequences of the molecular adaptation of LTR5HS for enhancer function have been beneficial to the host, it is nonetheless tempting to speculate that some LTR5HS-driven changes in gene expression may have measurable phenotypic consequences on early development, endowing it with ape-specific attributes. Regardless, the CARGO-CRISPRa/CRISPRi strategy described here provides a novel tool to study the impact of LTRs and other TEs on primate-specific features of development and disease.

## Materials and methods

| Reagent type (species) or resource | Designation | Source or reference | Identifiers | Additional information |
|---|---|---|---|---|
| Cell line (*H. sapiens*) | NCCIT | ATCC | ATCC:CRL-2073; RRID:CVCL_1451 | |
| Transfected construct (*H. sapiens*) | NCCIT PiggyBac dCas9-VPR | this paper | | Progenitors: NCCIT, PiggyBac transposon |
| Transfected construct (*H. sapiens*) | NCCIT PiggyBac dCas9-KRAB | this paper | | Progenitors: NCCIT, PiggyBac transposon |

*Continued on next page*

*Continued*

| Reagent type (species) or resource | Designation | Source or reference | Identifiers | Additional information |
|---|---|---|---|---|
| Transfected construct (*H. sapiens*) | NCCIT PiggyBac dCas9-GFP | this paper | | Progenitors: NCCIT, PiggyBac transposon |
| Transfected construct (*H. sapiens*) | NCCIT PiggyBac dCas9-VPR LTR5HS *S. pyogenes* | this paper | | Progenitor: NCCIT PiggyBac dCas9-VPR |
| Transfected construct (*H. sapiens*) | NCCIT PiggyBac dCas9-VPR LTR5HS *S. aureus* | this paper | | Progenitor: NCCIT PiggyBac dCas9-VPR |
| Transfected construct (*H. sapiens*) | NCCIT PiggyBac dCas9-VPR nontarget *S. pyogenes* | this paper | | Progenitor: NCCIT PiggyBac dCas9-VPR |
| Transfected construct (*H. sapiens*) | NCCIT PiggyBac dCas9-KRAB LTR5HS *S. pyogenes* | this paper | | Progenitor: NCCIT PiggyBac dCas9-KRAB |
| Transfected construct (*H. sapiens*) | NCCIT PiggyBac dCas9-KRAB LTR5HS *S. aureus* | this paper | | Progenitor: NCCIT PiggyBac dCas9-KRAB |
| Transfected construct (*H. sapiens*) | NCCIT PiggyBac dCas9-KRAB nontarget *S. pyogenes* | this paper | | Progenitor: NCCIT PiggyBac dCas9-KRAB |
| Transfected construct (*H. sapiens*) | NCCIT PiggyBac dCas9-GFP LTR5HS *S. pyogenes* | this paper | | Progenitor: NCCIT PiggyBac dCas9-GFP |
| Transfected construct (*H. sapiens*) | NCCIT PiggyBac dCas9-GFP LTR5HS *S. aureus* | this paper | | Progenitor: NCCIT PiggyBac dCas9-GFP |
| Transfected construct (*H. sapiens*) | NCCIT PiggyBac dCas9-GFP nontarget *S. pyogenes* | this paper | | Progenitor: NCCIT PiggyBac dCas9-GFP |
| Antibody | HERVK env | Austral Biologicals | Austral Biologicals: HERM-1811–5 | See *Supplementary file 2* |
| Antibody | RNA pol II (clone 8WG16) | Biolegend | Biolegend:920101; RRID:AB_2565317 | See *Supplementary file 2* |
| Antibody | Cas9 (clone 8C1-F10) | Active Motif | Active Motif:61757 | See *Supplementary file 2* |
| Antibody | GFP | Abcam | Abcam:ab290; RRID:AB_303395 | See *Supplementary file 2* |
| Antibody | GFP | Thermo Fisher Scientific (Invitrogen) | Thermo Fisher Scientific (Invitrogen): A-11122; RRID:AB_221569 | See *Supplementary file 2* |
| Antibody | H3K27ac | Active Motif | Active Motif:39133; RRID:AB_2561016 | See *Supplementary file 2* |
| Antibody | H3K4me3 | Active Motif | Active Motif:39159; RRID:AB_2615077 | See *Supplementary file 2* |
| Antibody | H3K9me3 | Abcam | Abcam:ab8898; RRID:AB_306848 | See *Supplementary file 2* |
| Recombinant DNA reagent | PiggyBac transposon | System Biosciences | | |
| Recombinant DNA reagent | px332 | PMID: 29371426 | | |
| Recombinant DNA reagent | LTR5HS *S. pyogenes* scaffold CARGO array | this paper | | Progenitors: PiggyBac transposon, px332; targeting array (LTR5HS gRNAs) |

*Continued on next page*

*Continued*

| Reagent type (species) or resource | Designation | Source or reference | Identifiers | Additional information |
|---|---|---|---|---|
| Recombinant DNA reagent | LTR5HS *S. aureus* scaffold CARGO array | this paper | | Progenitors: PiggyBac transposon, px332; control array (LTR5HS gRNAs) |
| Recombinant DNA reagent | nontarget *S. pyogenes* scaffold CARGO array | this paper | | Progenitors: PiggyBac transposon, px332; control array (nontargeting gRNAs) |
| Recombinant DNA reagent | px458 GFP | PMID: 24157548 | Addgene:48138 | |
| Recombinant DNA reagent | px458 mCherry | this paper | | Progenitor: px458 GFP |
| Recombinant DNA reagent | PiggyBac dCas9-VPR | this paper | | Progenitor: PiggyBac transposon |
| Recombinant DNA reagent | PiggyBac dCas9-KRAB | this paper | | Progenitor: PiggyBac transposon |
| Recombinant DNA reagent | PiggyBac dCas9-GFP | this paper | | Progenitor: PiggyBac transposon |
| Sequence-based reagent | RT-qPCR primers (*Supplementary file 2*) | this paper | | |
| Sequence-based reagent | ChIP-qPCR primers (*Supplementary file 2*) | this paper | | |
| Sequence-based reagent | CARGO CRISPR gRNAs (*Supplementary file 2*) | this paper | | |
| Sequence-based reagent | LTR5HS deletion CRISPR gRNAs (*Supplementary file 2*) | this paper | | |
| Commercial assay or kit | Lonza MycoAlert | Lonza | Lonza:LT07-418 | |
| Chemical compound, drug | Doxycycline hyclate | Sigma-Aldrich | Sigma-Aldrich:D9891 | |
| Chemical compound, drug | Puromycin | InvivoGen | InvivoGen:ant-pr-1 | |
| Chemical compound, drug | G418 | Thermo Fisher Scientific | Thermo Fisher Scientific:10131–035 | |
| Software, algorithm | CRISPOR | PMID: 27380939 | | |
| Software, algorithm | Bowtie | PMID: 19261174 | | |
| Software, algorithm | Bedtools | PMID: 25199790 | | |
| Software, algorithm | FastQC | Other | | https://www.bioinformatics.babraham.ac.uk/projects/fastqc/ |
| Software, algorithm | Bowtie2 | PMID: 22388286 | | |
| Software, algorithm | Samtools | PMID: 19505943 | | |
| Software, algorithm | Picard tools | Other | | https://broadinstitute.github.io/picard/ |
| Software, algorithm | macs2 | PMID: 18798982 | | |
| Software, algorithm | Deeptools | PMID: 27079975 | | |

*Continued*

| Reagent type (species) or resource | Designation | Source or reference | Identifiers | Additional information |
|---|---|---|---|---|
| Software, algorithm | HOMER | PMID: 20513432 | | |
| Software, algorithm | cutadapt | Other | | https://github.com/marcelm/cutadapt |
| Software, algorithm | hisat2 | PMID: 25751142 | | |
| Software, algorithm | featurecounts | PMID: 24227677 | | |
| Software, algorithm | DESeq2 | PMID: 25516281 | | |
| Software, algorithm | Tophat2 | PMID: 23618408 | | |
| Software, algorithm | skewer | PMID: 24925680 | | |
| Software, algorithm | StringTie | PMID: 25690850 | | |

## Cell culture

NCCIT cells were obtained from ATCC. NCCIT cells were grown in RPMI-1640 (Thermo Fisher Scientific, Waltham, MA, USA), supplemented with 10% FBS (Omega Scientific, Tarzana, CA, USA), 1x Glutamax (Thermo Fisher Scientific), 1x non-essential amino acids (Thermo Fisher Scientific), and 1x antibiotic/antimycotic (Thermo Fisher Scientific). Cell lines were tested for mycoplasma contamination using MycoAlert Detection Kit (Lonza, Basel, Switzerland). All cell lines tested negative for mycoplasma contamination.

## LTR5HS-targeting CRISPR gRNA design

All unique SpCas9 16 nt seed sequences derived from known instances of LTR5HS were aligned against hg38 human genome with bowtie (*Langmead et al., 2009*) using '-v' mode with up to three mismatches allowed. Alignments to LTR5HS, LTR5A and LTR5B were not counted as off-targets. The twelve guides with lowest off-target rate were selected for the targeting array. Non-targeting guides were taken from (*Shalem et al., 2014*).

## LTR5HS gRNA analysis

For analysis shown in *Figure 1—figure supplement 1*, to identify potential binding sites for LTR5HS-targeting gRNAs in silico, Repeatmasker table was downloaded from UCSC table browser hg38, converted to BED format, and subsetted for specific analyses. Specifically, records for LTR5HS, LTR5A, and LTR5B were extracted into separate BED files, then FASTA files for these files were extracted using bedtools getfasta function (*Quinlan, 2014*). Bowtie indices were built for these FASTA files, and the set of LTR5HS-targeting gRNAs was aligned to each LTR5x index allowing 0, 1, 2, or 3 mismatches, with 'bowtie -S -f -a -v {0, 1, 2, or 3} $index guides.fa > aligned.sam' used as the exact command. SAM files were converted to BED using bedtools bamtobed function, then the PAM sequence for each alignment was extracted using bedtools getfasta function. Only guide alignments followed by the PAM sequence 'NGG' were counted.

## CARGO assembly

CARGO arrays containing twelve guides were assembled as described previously (*Gu et al., 2018*). The 12 gRNA transcriptional units of the CARGO plasmid were inserted into a PiggyBac transposon plasmid (System Biosciences, Palo Alto, CA, USA) containing a neomycin-selectable cassette by traditional cloning.

## Plasmids

For CRISPRa/CRISPRi, dCas9-VPR, dCas9-KRAB, and dCas9-GFP fusions were inserted into a Piggy-Bac transposon containing a puromycin-selectable cassette. For CRISPR/Cas9 deletion of LTR5HS, a guide upstream of a targeted LTR5HS insertion was cloned into px458 (pSpCas9(BB)−2A-GFP, a gift from Feng Zhang, Addgene plasmid #48138) (Ran et al., 2013), which expresses Cas9 and GFP, and a guide downstream of a targeted LTR5HS insertion was cloned into a modified px458 plasmid which expresses mCherry instead of GFP.

## Generation of stable lines

NCCIT cells were transfected with PiggyBac plasmids containing a dox-inducible dCas9 fusion, along with PiggyBac transposase, and selected using puromycin (Invivogen, San Diego, CA, USA). These lines were then transfected with PiggyBac CARGO plasmids, along with PiggyBac transposase, and selected using G418 (Thermo Fisher Scientific). Cells were re-selected with puromycin (Invivogen) to ensure that dCas9-fusions were not lost during second transposition event. For all dCas9 fusion experiments, expression of fusion proteins was induced for four days with 2 ug/mL doxycycline (Sigma-Aldrich, St. Louis, MO, USA)

## RNA extraction

Cells for RT-qPCR and RNA-seq were homogenized in Trizol (Thermo Fisher Scientific), then RNA was extracted using Direct-zol RNA columns (Zymo Research, Irvine, CA, USA), with DNase treatment on-column, and eluted in water.

## Reverse transcription for RT-qPCR

Reverse transcription for RT-qPCR was performed using SensiFAST cDNA synthesis kit (Bioline, Taunton, MA, USA), according to the manufacturer's instructions with input from the RNA extraction described above.

## qPCR

qPCR was performed using SensiFAST SYBR No-Rox kit (Bioline) in a LightCycler 480II (Roche, Basel, Switzerland), using technical duplicates or triplicates for each sample. Each condition was also analyzed with at least two independent biological replicates. Figure legends indicate transcript normalization for RT-qPCR.

## Protein extraction and western blotting

Whole cell nuclear extracts were prepared by lysing cells for 30 min at 4° C with overhead vertical rotation in protein extraction buffer (300 mM NaCl, 100 mM Tris pH 8, 0.2 mM EDTA, 0.1% Triton X-100, 10% glycerol, with 1x cOmplete EDTA-free protease-inhibitor cocktail [Roche]), then clearing by centrifugation and recovery of the supernatant. Total protein concentration was quantified by Bradford assay (Bio-Rad, Hercules, CA, USA). Equal amounts of protein were denatured in LDS buffer (Thermo Fisher Scientific) supplemented with 2-mercaptoethanol (Sigma-Aldrich), then loaded in 3-fold serial dilutions onto tris-glycine 4–20% SDS-PAGE denaturing gradient gels (Thermo Fisher Scientific), then transferred onto nitrocellulose membrane. Chemiluminescence was assayed using Lumi-light Plus (Roche) or Amersham (GE Life Sciences, Pittsburgh, PA, USA) and visualized on autoradiography film.

## Chromatin immunoprecipitation

ChIP assays were performed as described previously (Rada-Iglesias et al., 2011). Briefly, approximately $10^7$ NCCIT cells were fixed in 1% formaldehyde for 10 min at room temperature in PBS, then quenched with glycine to a final concentration of 0.125 M for 10 min. Chromatin was sonicated to 0.5–2.0 kb using Bioruptor (Diagenode, Liège, Belgium), cleared by centrifugation, divided into separate aliquots for each antibody, and incubated with 5 µg of antibody overnight at 4° C. Subsequently, 100 µL of Dynabeads protein G (Thermo Fisher Scientific) were added to the ChIP reactions and incubated for 4–6 hr at 4° C. Magnetic beads were washed and chromatin was eluted, followed by reversal of crosslinks overnight at 65° C, proteinase K and RNase A treatment, and DNA

purification by phenol/chloroform/isoamyl alcohol extraction and ethanol precipitation. ChIP DNA was resuspended in water.

## Library preparation and sequencing for ChIP-seq

For ChIP-seq data presented in *Figure 2* and its supplements, ChIP DNA (10 ng) was end-repaired, A-tailed, and ligated to NEBNext adapter for Illumina (New England Biolabs, Ipswich, MA, USA), followed by cleavage with USER enzyme (New England Biolabs). Adapter-ligated DNA was size-selected using a left/right AMPure XP size selection (Beckman Coulter, Brea, CA, USA). Size-selected DNA was amplified by qPCR using one universal primer and one indexed primer (New England Biolabs), then cleaned up with two AMPure XP cleanups. For ChIP-seq data presented in *Figure 3* and in *Figure 5E* and *Figure 5—figure supplement 3*, libraries were prepared using Ovation Ultralow System V2 UDI (NuGEN Technologies, San Carlos, CA, USA) according to manufacturer's instructions, starting with 10 ng of ChIP DNA. Library DNA was analyzed on Bioanalyzer DNA HS (Agilent, Santa Clara, CA, USA), then pooled and sequenced on a HiSeq 4000 (Illumina, San Diego, CA, USA) at the Stanford Genome Sequencing Service Center, using 2 × 150 bp sequencing with index read or dual index read.

## Library preparation and sequencing for RNA-seq

Total RNA (10 ug) from two independent biological replicates was subjected to oligo-dT purification using Dynabeads oligo(dT) (Thermo Fisher Scientific), then fragmented with 10x fragmentation buffer (Thermo Fisher Scientific). Fragmented RNA was used for first strand cDNA synthesis with Superscript II (Thermo Fisher Scientific) and random hexamer primers (Thermo Fisher Scientific). Second strand cDNA synthesis was performed using RNase H (Thermo Fisher scientific) and DNA polymerase I (New England Biolabs). The resulting double-stranded cDNA was used for Illumina library preparation as described for ChIP-seq experiments, but was size-selected on acrylamide gels, and pooled and sequenced on a NextSeq 500 (Illumina) at the Stanford Functional Genomics Facility, using 2 × 150 sequencing with index read.

## CRISPR/Cas9 deletion of LTR5HS

gRNAs upstream and downstream of individual LTR5HS insertions with low potential off-targets were identified using CRISPOR (*Haeussler et al., 2016*). For each deletion, two guides were selected, one upstream and one downstream of the LTR5HS. To avoid deletion of multiple LTR5HS, guides were chosen that do not overlap the LTR5HS. NCCIT cells were transfected with Lipofect-amine 2000 (Thermo Fisher Scientific) with px458-GFP and px458-mCherry plasmids containing upstream and downstream gRNAs for a single LTR5HS insertion. 48 hr later, 1500 GFP- and mCherry- dual-fluorescent cells were sorted on a FACSAria II (BD Biosciences, San Jose, CA, USA), then plated onto a single well of a 6-well plate, coated with 10 ug/mL human plasma fibronectin (MilliporeSigma, Burlington, MA, USA). After ~5–7 days, individual colonies derived from single cells were picked and plated onto a single well of a 96-well fibronectin-coated plate. Cells were grown to confluency, then passaged, genotyped with DirectPCR Lysis Reagent (Viagen Biotech, Los Angeles, CA, USA) by PCR, and analyzed for gene expression by RT-qPCR. Multiple deletion and wild type clones for each LTR5HS insertion were analyzed, as indicated in *Figure 6B*. Each clone was analyzed at two separate passages.

## ChIP-seq analysis

Quality of FASTQ files was assessed using FASTQC software. Reads were aligned to hg38 genome using bowtie2 (*Langmead and Salzberg, 2012*), with 'bowtie2 -p $threads –end-to-end –no-mixed –no-discordant –minins 100 –maxins 1000 -x hg38 −1 $read1 −2 $read2 > aligned.sam' as the exact command for each sample. SAM files were converted to sorted, indexed, compressed BAM files using SAMtools (*Li et al., 2009*). Duplicate reads were removed using the MarkDuplicates function of Picard Tools. Macs2 (*Zhang et al., 2008*) callpeak function was used to call peaks for each ChIP (condition/antibody combination). For each ChIP with each antibody, peaks were called using that ChIP as the 'treatment' and the other two condition ChIPs as the 'control' for macs2, as previous studies have used other ChIP samples, Cas9 alone ChIPs, or ChIPs from cells not expressing Cas9 as controls (*Kuscu et al., 2014*; *Polstein et al., 2015*; *Wu et al., 2014*). Overlaps between ChIP peak

calls were performed using bedtools intersect function. Deeptools command line tools (*Ramírez et al., 2016*) was used to generate Bigwig plots for visualization of UCSC genome browser and ChIP-seq heat maps. HOMER software (*Heinz et al., 2010*) was used to associate dCas9 ChIP-seq peaks to different genomic features.

## ChIP-seq to gRNA alignments correlation analysis

For analysis shown in *Figure 2—figure supplement 3*, the BED file of all LTR5HS insertions was intersected with BED file containing three-antibody overlap ChIP-seq peak calls for LTR5HS Sp condition. Each LTR5HS insertion, along with the number of gRNAs expected to align to it (at 0, 1, 2, or 3 mismatches allowed), was therefore matched to the macs2 ChIP score at the same LTR5HS, and these are plotted as a violin point plot using the vpplot function of the vipor R package. Spearman correlation coefficient ρ is reported in the text.

## RNA-seq analysis

Quality of FASTQ files was assessed using FASTQC software. Reads from were trimmed of Illumina adapter sequences using cutadapt. For analysis of human non-repeat transcripts, trimmed reads were aligned using hisat2 (*Kim et al., 2015*) to the hg38_tran index, with 'hisat2 -q -p $threads -t –no-mixed –no-discordant -x hg38_tran −1 $read1 −2 $read2 -S aligned.sam' as the exact command for each sample. Reads were assigned to gene models using featureCounts (*Liao et al., 2014*), and differential expression analysis was performed using DESeq2 (*Love et al., 2014*). For analysis of Repeatmasker transcripts, trimmed reads were aligned using TopHat2 (*Kim et al., 2013*) to an index built from a FASTA file containing all Repeatmasker sequences, which was itself built using bedtools getfasta command with the Repeatmasker BED file described above. Reads were assigned to repeat models using featureCounts, then RPKM was calculated from these tabulations. For comparison of early embryo single cell RNA-seq, rhesus reads from (*Wang et al., 2017*) were aligned to rheMac8, and human reads from (*Yan et al., 2013*) were aligned to hg38 using hisat2, then reads were assigned to gene models using featureCounts. Ensembl BioMart was used to identify only genes with one-to-one orthology between the two species, and only these were used for further analyses. Transcripts per million (TPM) was calculated for each gene at each stage in each species. For chimeric transcript identification, RNA-seq reads were trimmed with skewer (*Jiang et al., 2014*) and aligned to GRCh38_p7 assembly with hisat2 with the following settings: –dta –no-mixed –no-discordant. Transcript models were built based on this alignment with StringTie (*Pertea et al., 2015*) and annotated with gffcompare using gencode25 transcript models. Spliced transcripts originating in or within 100 bp of LTR5HS were treated as chimeric transcripts. TPM corresponding to expression level of the known and new transcripts were calculated with separate StringTie run for each library alignment (stringtie -e -B -A).

## Unique mappability to LTR5HS

All possible 150 bp paired-end reads for fragments in size range 150–400 bp within −400 bp to +400 bp of known LTR5HS were generated from hg38 reference sequence with bedtools getfasta and aligned to hg38 assembly with bowtie2 (–end-to-end –no-mixed –no-discordant). MAPQ score for each pair was extracted and assigned to LTR5HS instance. Plot of fraction of uniquely mappable (MAPQ > 20) reads was generated in R.

## Antibodies, primers, gRNAs

All antibodies, primers, and gRNAs used in this study are listed in *Supplementary file 2*.

## Data availability

Sequencing data have been deposited in GEO under accession code GSE111337.

## Acknowledgements

We thank B Gu for initial discussions on the design of the CARGO arrays, and R Srinivasan and M Bauer for assistance with cloning. We also thank E Grow and members of the Wysocka lab for helpful comments on this manuscript. Finally, we thank the Stanford Genome Sequencing Service Center by

the Stanford Center for Genomics and Personalized Medicine (supported by the NIH grant S10OD020141) and John Coller and the staff of the Stanford Functional Genomics Facility for sequencing services.

## Additional information

### Funding

| Funder | Grant reference number | Author |
|---|---|---|
| National Science Foundation | Graduate Research Fellowship Program | Daniel R Fuentes |
| Howard Hughes Medical Institute | | Joanna Wysocka |
| National Institute of General Medical Sciences | R01GM112720 | Joanna Wysocka |

The funders had no role in study design, data collection and interpretation, or the decision to submit the work for publication.

### Author contributions

Daniel R Fuentes, Conceptualization, Data curation, Software, Formal analysis, Validation, Investigation, Visualization, Methodology, Writing—original draft, Writing—review and editing; Tomek Swigut, Conceptualization, Software, Formal analysis, Validation, Visualization, Methodology; Joanna Wysocka, Conceptualization, Supervision, Funding acquisition, Validation, Methodology, Writing—original draft, Project administration, Writing—review and editing

### Author ORCIDs

Daniel R Fuentes (iD) http://orcid.org/0000-0002-0412-6933
Tomek Swigut (iD) http://orcid.org/0000-0002-7649-6781
Joanna Wysocka (iD) http://orcid.org/0000-0002-6909-6544

### Decision letter and Author response

Decision letter https://doi.org/10.7554/eLife.35989.042
Author response https://doi.org/10.7554/eLife.35989.043

## Additional files

### Supplementary files

• Supplementary file 1. Excel file of statistical analysis for *Figure 5*
DOI: https://doi.org/10.7554/eLife.35989.018

• Supplementary file 2. Excel file of antibodies, primers, and gRNAs used in this study
DOI: https://doi.org/10.7554/eLife.35989.019

• Supplementary file 3. Text file of analyzed RNA-seq data generated in this study. Includes gene name; log2 fold change and adjusted p-value for CRISPRa and CRISPRi; hg38 coordinates of the nearest LTR5HS insertion to the TSS of the gene; and distance between the TSS and the LTR5HS.
DOI: https://doi.org/10.7554/eLife.35989.020

• Supplementary file 4. Text file of analyzed RNA-seq data from (*Wang et al., 2017*; *Yan et al., 2013*). Includes gene name (for all 15090 genes with one-to-one orthology between human and rhesus); Boolean field indicating whether the gene is one of the 193 LTR5HS-regulated transcripts with one-to-one-orthology, which are plotted in *Figure 4E*; TPM values for oocyte, zygote, 2-cell, 4-cell, and 8-cell stages, morula, and blastocyst of both human and rhesus.
DOI: https://doi.org/10.7554/eLife.35989.021

• Supplementary file 5. BED of dCas9 ChIP-seq peaks for LTR5HS *S. pyogenes* (i.e. targeting) condition in *Figure 2*. Includes hg38 coordinates and MACS2 score for each peak.
DOI: https://doi.org/10.7554/eLife.35989.022

• Transparent reporting form
DOI: https://doi.org/10.7554/eLife.35989.023

## Data availability

Sequencing data have been deposited in GEO under accession code GSE111337.

The following datasets were generated:

| Author(s) | Year | Dataset title | Dataset URL | Database, license, and accessibility information |
|---|---|---|---|---|
| Fuentes DR, Swigut T, Wysocka J | 2018 | Systematic perturbation of retroviral LTRs reveals widespread long-range effects on human gene regulation [ChIP-seq] | https://www.ncbi.nlm.nih.gov/geo/query/acc.cgi?acc=GSE111331 | ublicly available at the NCBI Gene Expression Omnibus (accession no: GSE111331) |
| Fuentes DR, Swigut T, Wysocka J | 2018 | Systematic perturbation of retroviral LTRs reveals widespread long-range effects on human gene regulation [RNA-seq] | https://www.ncbi.nlm.nih.gov/geo/query/acc.cgi?acc=GSE111332 | Publicly available at the NCBI Gene Expression Omnibus (accession no: GSE111332) |
| Fuentes DR, Swigut T, Wysocka J | 2018 | Systematic perturbation of retroviral LTRs reveals widespread long-range effects on human gene regulation | https://www.ncbi.nlm.nih.gov/geo/query/acc.cgi?acc=GSE111337 | Publicly available at the NCBI Gene Expression Omnibus (accession no: GSE111337) |

The following previously published datasets were used:

| Author(s) | Year | Dataset title | Dataset URL | Database, license, and accessibility information |
|---|---|---|---|---|
| Wang X, Liu D | 2017 | Transcriptome analyses of rhesus monkey pre-implantation embryos reveal a reduced capacity for DNA double strand break (DSB) repair in primate oocytes and early embryos | https://www.ncbi.nlm.nih.gov/geo/query/acc.cgi?acc=GSE86938 | Publicly available at the NCBI Gene Expression Omnibus (accession no: GSE86938) |
| Ji X | 2015 | 3D Chromosome Regulatory Landscape of Human Pluripotent Cells [ChIP-Seq] | https://www.ncbi.nlm.nih.gov/geo/query/acc.cgi?acc=GSE69646 | Publicly available at the NCBI Gene Expression Omnibus (accession no: GSE69646) |
| Tang F, Qiao J, Li R | 2013 | Tracing pluripotency of human early embryos and embryonic stem cells by single cell RNA-seq | https://www.ncbi.nlm.nih.gov/geo/query/acc.cgi?acc=GSE36552 | Publicly available at the NCBI Gene Expression Omnibus (accession no: GSE36552) |
| Jones P | 2013 | Genome-wide maps of chromatin remodeler SNF5 in human pluripotent cells | https://www.ncbi.nlm.nih.gov/geo/query/acc.cgi?acc=GSE36134 | Publicly available at the NCBI Gene Expression Omnibus (accession no: GSE36134) |
| Takashima Y, Guo G, Loos R, Nichols J, Ficz G, Krueger F, Oxley D, Santos F, Clarke J, Mansfield W, Reik W, Bertone P, Smith A | 2014 | RNA sequencing of conventional and reset human pluripotent stem cells | https://www.ebi.ac.uk/arrayexpress/experiments/E-MTAB-2857/ | Publicly available at the EMBL-EBI Array Express archive (accession no. E-MTAB-2857) |
| Theunissen TW, Friedli M, He Y, Planet E, Oneil R, Markoulaki S, Wang H, Pontis J, Iouranova A, Imbeault M, Duc J, Cohen M, Wert KJ, Cas- | 2016 | Molecular Criteria for Defining the Naive Human Pluripotent State | https://www.ncbi.nlm.nih.gov/geo/query/acc.cgi?acc=GSE75868 | Publicly available at the NCBI Gene Expression Omnibus (accession no: GSE75868) |

tanon RG, Zhang Z,
Maetzel D, Huang
Y, Nery JR, Drotar
J, Lungjangwa T,
Trono D, Ecker JR,
Jaenisch R

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
