## [Decision Letter]

Thank you for submitting your article "Systematic perturbation of retroviral LTRs reveals widespread long-range effects on human gene regulation" for consideration by *eLife*. Your article has been reviewed by three peer reviewers, and the evaluation has been overseen by a Reviewing Editor and Detlef Weigel as the Senior Editor. The following individuals involved in review of your submission have agreed to reveal their identity: Deborah Bourchis (Reviewer #1); Cédric Feschotte (Reviewer #2).

The reviewers have discussed the reviews with one another and the Reviewing Editor has drafted this decision to help you prepare a revised submission.

Summary:

In this study, the question of the extent to which Transposable Elements and their relics in a genome have been adapted to influence "host" gene regulation is addressed. By adapting their recently published gRNA multiplexing CARGO method (Gu et al., 2018) the authors target dCAS9-fusion activator (VPR) or repressor (KRAB) proteins to the ~700 copies of the HERVK (HML-2), LTR5HS elements of the human genome (known to be expressed early on in human development). The authors examine the effects of this TE modulation on nearby gene expression in human embryonal carcinoma cells derived from a germ-line tumor (NCCIT). The authors report that activation/silencing of LTR5HS is associated with reciprocal up- and down-regulation of nearly 300 human genes, although the VPR activation shows more striking effects than KRAB repression. They go on to look at the range of impact that these LTR5HS elements have is up to 160kb, suggesting that they might be acting as potential enhancers. The authors also specifically delete three individual LTR5HS elements, and examine effects on nearby genes by qRT-PCR which points to their potential role in gene regulation.

This study is of a great interest for the field as it serves as an important demonstration that expression of hundreds of copies of a specific family of LTR retrotransposons can be efficiently modulated using dCas9 effectors and the recently developed CARGO system and it allows the impact of a whole LTR family on gene expression to be addressed in a systematic way. Although previous studies (Guallar et al.,; Ishiuchi et al.,; Jachowicz et al., 2017; Amabile et al., 2016 – which should have been cited) have targeted whole TE families, the present study is one of the first to describe an in depth, genome-wide analysis of what the functional contribution of LTR elements might be for gene expression control across the human genome. The manuscript is well written, easy to read and the experiments are of high quality. There are however some important issues that the authors will need to address.

Essential revisions:

1) The authors claim that the LTR5HS sequences they have affected are acting as enhancers. However, they need to prove this enhancer potential at the endogenous locations. As the study stands they cannot rule out that the elements are acting as alternative promoters, particularly as the three LTR5HS elements that they deleted were in close proximity (a few kb) to the associated genes. Several approaches can address this, as outlined in the reviewers' comments. These include demonstrating enhancer orientation independence for the LTRs; presenting their analyses in Figure 4 using stranded information, and directionality of the LTR as well as clarifying distances from TSS; demonstrating that chimeric transcripts are not produced between the LTR5HS elements and the up-regulated genes in the CRISPRa approach using 5'RACE or CAGE RNAseq; performing ChIP-PCR to examine enhancer marks such as H3K27ac, H3K122ac and H3K4me1 to see if their distributions change in wt versus LTR-deleted condition across at least two of the loci studied (enhancer, promoter, and LTR); performing deletion experiments for LTRs that are more distant to the misregulated genes identified; examining the effects both upstream and downstream of the genes.

2) The authors should provide more information about the transcription factor and chromatin landscape of the LTR sequences. This could thus assess whether there is a pre-existing regulatory activity of individual LTR5HS elements in NCCIT cells that could influence the efficiency and outcomes of the CRISPRi/a assays.

3) The repressive effects on neighbouring genes upon KRAB targeting is not clear (Figure 4B and supplementary figures). There are no statistics and the number of genes falling into different categories are not given; the effects on basal transciprition of CACNA2D2, *NFKB2* and SERPINB9 upon deletion of the unique LTR is not shown.

4) Statistics need to be checked and included in several of the figures including Figure 4 above. The distance analyses lack statistics: how many genes per 'distance' block were analysed? Are the changes in transcription statistically significant? How would this look with a negative control (e.g. another repeat)?

---

## [Author Response]

Essential revisions:1) The authors claim that the LTR5HS sequences they have affected are acting as enhancers. However, they need to prove this enhancer potential at the endogenous locations. As the study stands they cannot rule out that the elements are acting as alternative promoters, particularly as the three LTR5HS elements that they deleted were in close proximity (a few kb) to the associated genes. Several approaches can address this, as outlined in the reviewers' comments. These include demonstrating enhancer orientation independence for the LTRs; presenting their analyses in Figure 4 using stranded information, and directionality of the LTR as well as clarifying distances from TSS; demonstrating that chimeric transcripts are not produced between the LTR5HS elements and the up-regulated genes in the CRISPRa approach using 5'RACE or CAGE RNAseq; performing ChIP-PCR to examine enhancer marks such as H3K27ac, H3K122ac and H3K4me1 to see if their distributions change in wt versus LTR-deleted condition across at least two of the loci studied (enhancer, promoter, and LTR); performing deletion experiments for LTRs that are more distant to the misregulated genes identified; examining the effects both upstream and downstream of the genes.

We thank the reviewers for this comment and agree that the claim that LTR5HS elements function as distal enhancers is central to the thesis of our manuscript, and thus should be further strengthened. We followed the reviewers’ suggestions and took a multi-pronged approach to addressing this comment, as outlined below:

Demonstrating orientation independence for the LTR5HS effect on gene expression: We reanalyzed the data concerning the LTR5HS-regulated genes (as defined in Figure 4A of the revised manuscript) and their nearest LTR5HS insertion (the putative enhancer). We found that for these 275 genes, the nearest LTR5HS is upstream of the promoter in 150 cases, and downstream in 125 cases. This finding suggests that even downstream LTR5HS insertions can have a transcriptional effect on the gene, a finding not compatible with their function as alternative promoters. Importantly, we deleted three of those downstream LTR5HS insertions and confirmed that they indeed significantly affect expression of the candidate target gene (see next section for details). As LTR sequences do have a natural orientation, we also examined the relative orientation of the nearest LTR5HS for each of these genes. In the 150 cases where the LTR5HS insertion is upstream of the promoter, the LTR5HS insertion has the same orientation (both on the Watson strand or both on the Crick strand) 83 times, compared to 67 in the opposite orientation. In the 125 cases where the LTR5HS insertion is downstream, the insertion has the same orientation as the gene 48 times, compared to 77 in the opposite orientation. These findings together suggest that neither the relative position of the LTR5HS to the promoter, nor its orientation, determines its ability to effect a transcriptional change on the gene in question under CRISPRa or CRISPRi, consistent with the proposed enhancer function. This information has now been incorporated in the text of the revised manuscript under the header “Reciprocal effects of LTR5HS CRISPRa/CRISPRi on host gene expression.”

Additional deletion experiments for LTRs that are more distant to the misregulated genes identified and located downstream from the TSS:

The reviewers suggested that the three LTR5HS elements deleted in our original manuscript were all *“in close proximity”* to the associated genes and thus may not be enhancers. First, we would like to clarify that since these LTRs were respectively ~17 kb, ~2 kb and ~6 kb upstream from the promoter, they all fall under the definition of distal regulatory elements, typically defined as those that do not overlap with promoter sequences (usually contained within 200-500 bp of the TSS). Furthermore, two of the genes (*SERPINB9* and *CACNA2D2*) are transcribed in opposite orientation relative to the LTR5HS, making their LTR5HS-originating chimeric transcription unlikely. Nonetheless, we agree that since all deleted elements were upstream from the target genes, additional deletions were needed to further exclude a possibility that the LTR5HS elements function as alternative promoters. In the revised manuscript, we present analysis of clonal lines with homozygous deletions of three LTR5HS elements located downstream from the candidate target genes: (i) ~16 kb downstream of the *ALPPL2* gene TSS, (ii) ~245 kb downstream of the *EPHA7* gene TSS (as this gene is long, the deletion is 65 kb downstream of the annotated transcription termination site [TTS]) and (iii) ~69 kb downstream of the *GDPD1* gene TSS. In all three cases, we observed downregulation of the candidate target gene (see Figure 6B in the revised manuscript). Together with our previous results, we now show examples of deletions of LTR5HS elements located within a wide distance range from the target gene promoter (e.g. from 2 kb for the closest to 245 kb for the most distal), positioned either upstream or downstream from the gene TSS and transcribed in either direction with respect to the gene. In all cases, we observed downregulation of the candidate target gene (Figure 6B), providing definitive evidence that LTR5HS elements indeed function as enhancers.

We also want to clarify (since there may have been some confusion, see essential revision point 3), that these deletion experiments were all performed in WT NCCIT cells that do not express any dCas9 fusions. We believe that the confusion might have been caused by our inclusion of the genome browser images with dCas9 binding at these selected LTRs in the figure, and thus we eliminated those browser images and reorganized the figure to incorporate new data.

ChIP-PCR analysis of enhancer marks in the LTR5HS deletion lines:

We performed ChIP-qPCR for the histone modifications H3K27ac and H3K4me1 in homozygous deletion and wild type clones for three separate LTR5HS insertions: those near *CACNA2D2, ALPPL2*, and *EPHA7*. In each case, we observe loss of both H3K27ac and H3K4me1 from the LTR5HS flanking regions in the deletion lines (shown in Figure 6A of the revised manuscript), demonstrating that deposition of these enhancer marks is dependent on the presence of the LTR sequence. Furthermore, analysis of the H3K27ac at the target gene promoters revealed downregulation of promoter acetylation levels in LTR5HS deletion lines, though we note that this downregulation did not reach statistical significance at one of the genes (Figure 6—figure supplement 1).

Analysis of chimeric transcripts between the LTR5HS elements and the upregulated genes: We analyzed deeply sequenced (~500 million 150 bp paired-end reads) RNA-seq data for the presence of the chimeric transcripts between LTR5HS and 275 LTR5HS-regulated genes identified in our study. We detected an appreciable level (e.g. > 1 TPM, transcript per million) of chimeric transcription at only four of the 275 genes (specifically, *NBPF12, SLC4A8, FA2H* and *TIMM50*). Notably, of the six LTR5HS elements that we deleted, all showed effect on candidate target gene expression, but none had detectable levels of chimeric transcription between the LTR5HS and the regulated gene. We therefore conclude that while such chimeric transcripts may indeed arise at some loci, they cannot explain the regulatory effects observed in our study. This information has been incorporated into the manuscript under the header, “Reciprocal effects of LTR5HS CRISPRa/CRISPRi on host gene expression.”

2) The authors should provide more information about the transcription factor and chromatin landscape of the LTR sequences. This could thus assess whether there is a pre-existing regulatory activity of individual LTR5HS elements in NCCIT cells that could influence the efficiency and outcomes of the CRISPRi/a assays.

To investigate the chromatin landscape of the LTR sequences, we performed ChIP-seq for H3K27ac, H3K4me3, and H3K9me3. In addition to doing these experiments in the parental WT NCCIT line, we also performed these ChIPs in lines expressing dCas9-VPR and dCas9-KRAB along with the LTR5HS *S. pyogenes* CARGO array (i.e. under targeting conditions). This experimental design allows us to not only examine the landscape in unperturbed NCCITs, but also to measure the effects of VPR activation and KRAB repression on the chromatin states. Results from these experiments are presented in a new figure (Figure 3 in the revised manuscript, subsequent figures have been renamed accordingly).

Briefly, our results can be summarized as follows:

H3K27ac, H3K4me3 and H3K9me3 patterns

A subset of LTR5HS elements is marked by H3K27ac and H3K4me3 in WT cells in the absence of perturbation, with H3K4me3 showing asymmetric distribution consistent with the direction of the LTR-driven transcription, as has previously been observed at promoters and highly transcribed enhancers. Under CRISPRa conditions, most (over 90%) LTR5HS elements gain high level of H3K27 acetylation, but interestingly, H3K4me3 levels remain relatively unaffected. Strong gains of H3K27ac occur even at those LTR5HS elements that have low/no endogenous acetylation, which may indicate that ectopic enhancer activation is relatively common and efficient with dCas9-VPR system. Conversely, under CRISPRi conditions, endogenous H3K27ac and H3K4me3 are suppressed, and most LTR5HS elements become decorated with high levels of H3K9me3, as would be expected, given that KRAB repression is mediated by the H3K9me3 deposition (Figure 3). Notably, in WT NCCIT cells, LTR5HS elements typically lack H3K9me3, regardless of the presence or absence of the active marks, suggesting that in these cells LTR5HS escapes KRAB-mediated repression, a major mechanism of endogenous retrovirus silencing.

In addition to examining histone modifications at LTR5HS insertions, we also assessed H3K27ac, H3K4me3, and H3K9me3 patterns surrounding the promoters of the 275 LTR5HSregulated transcripts (i.e. the genes activated by CRISPRa and repressed by CRISPRi, as defined in Figure 4A). We found that most of these promoters have at least some H3K27ac and H3K4me3 in WT cells, and most gain or lose, respectively, H3K27 acetylation under CRISPRa or CRISPRi conditions (new Figure 5E). Notably, these changes occur in the absence of direct dCas9 binding to the promoters, suggesting that they result from the long-range effects we describe. Furthermore, although some gains of H3K9me3 can be observed in the vicinity of the promoters under CRISPRi conditions, most of the TSS remain unmethylated at H3K9 and, unlike at the LTRs, their H3K4me3 levels are relatively unaffected, suggesting that direct silencing of promoters via H3K9me3 spreading from a nearby LTR5HS is not likely to explain the transcriptional effects we examine in this study. As a control, we performed these same analyses on a set of 275 randomly selected promoters, and we detected no changes in any histone mark under CRISPRa or CRISPRi conditions (Figure 5—figure supplement 3).

dCas9 binding patterns

We also performed ChIP-seq for dCas9-VPR and dCas9-KRAB fusions (in addition to the dCas9-GFP ChIP-seq reported in the original manuscript), as suggested in minor point 9. This experiment shows no dCas9 signal in WT (non-dCas9-expressing cells), but widespread binding to over 90% of LTR5HS in dCas9-VPR- and dCas9-KRAB-expressing cells (Figure 3A in the revised manuscript). We further observed that the dCas9-VPR levels at LTR5HS were higher than dCas9-KRAB levels. This is likely attributable to the fact that VPR, a strong activation domain, recruits coactivators that promote nucleosomal depletion, whereas KRAB-mediated H3K9me3 facilitates chromatin compaction, which may in turn provide, respectively, positive or negative feedback for dCas9 fusion binding. Nonetheless, dCas9-KRAB still occupies and mediates H3K9me3 deposition at the vast majority of LTR5HS elements (Figure 3A).

3) The repressive effects on neighbouring genes upon KRAB targeting is not clear (Figure 4B and supplementary figures). There are no statistics and the number of genes falling into different categories are not given; the effects on basal transciprition of CACNA2D2, NFKB2 and SERPINB9 upon deletion of the unique LTR is not shown.

A discussion of statistics in Figure 4B (now Figure 5B in revised manuscript) follows in point 4. As for the effects on basal transcription of *CACNA2D2, NFKB2*, and *SERPINB9* upon deletion of the nearest LTR5HS, this result was shown in the initial manuscript in Figure 5C (now Figure 6B in revised manuscript). Again, perhaps due to our inclusion of the genome browser images with dCas9 binding at these selected LTRs, it was not clear to the reviewers that these deletion experiments were all performed in WT NCCIT cells that do not express any dCas9 fusions. In any case, as discussed above, we now extended the analysis to the additional three elements and show the effect of deletion of six different LTR5HS on the basal transcript levels of their candidate target genes (Figure 6B).

4) Statistics need to be checked and included in several of the figures including Figure 4 above. The distance analyses lack statistics: how many genes per 'distance' block were analysed? Are the changes in transcription statistically significant? How would this look with a negative control (e.g. another repeat)?

(Please note that Figure 4 from the initial submission is now Figure 5 in the revised manuscript.)

We analyzed a total of 26517 genes. These are broken down into 11 bins (-200, -160, -120, -80, 40, 0, 40, 80, 120, 160, 200). Each bin contains genes whose TSS is within +/- 20 kb of the bin description. Therefore, bin “0” contains genes whose TSS are between -20 and +20 kb from the nearest LTR5HS; bin “40” contains genes whose TSS are between +20 and +60 kb from the nearest LTR5HS; and so on. The number of genes per bin is as follows:

BinNumber of genesCRISPRa Wilcoxon signedrank test P-valueCRISPRi Wilcoxon signedrank test P-value-2002279.29e-035.20e-02-1602707.33e-082.06e-04-1202575.37e-074.63e-06-802991.78e-186.80e-11-403215.63e-362.24e-2304084.37e-521.31e-47402687.30e-313.64e-22803063.46e-239.15e-141202543.66e-114.11e-061602604.70e-062.01e-022002314.73e-066.12e-04

As for statistical significance, we have now performed one-sample Wilcoxon signed rank tests of the null hypothesis that the distribution of log2 fold change at each bin is symmetric around 0, which would describe no change in gene expression. Those values are indicated in the table above. We would like to point out that all bins we show reach statistical significance, including the +/-220-180 kb bins. However, if we perform significance testing on further bins, we show that log2FoldChange figures do fail to disprove the null hypothesis of the Wilcoxon signed rank test at bins further away. See below for results:

Bin (kb)Number of genesCRISPRa Wilcoxon signedrank test PvalueCRISPRi Wilcoxon signedrank test P-value-420 to -3801780.2910.002-380 to -3402030.0200.775-340 to -3001710.7620.100-300 to -2601970.0350.801-260 to -2202190.0540.014220 to 2602180.0360.028260 to 3002410.0050.043300 to 3401970.8510.132340 to 3802000.2970.456380 to 4202080.4990.669

These results have now been incorporated to the revised manuscript as Supplemental file 1. To further address the reviewer’s point, as a negative control we generated similar plots of fold change under CRISPRa and CRISPRi, using an unrelated LTR, in this case LTR2 of the HERVE family, which has a similar number (~900) of insertions as LTR5HS, but is not active in NCCIT cells. It is clear that there is no effect at any distance (see Figure 5—figure supplement 1E-F in the revised manuscript).